# Gustatory interface for operative assessment and taste decoding in patients with tongue cancer

Xiner Wang[1,2,10], Guo Bai[3,10], Jizhi Liang[1,2,10], Qianyang Xie[3,10], Zhaohan Chen[4], Erda Zhou[1,2], Meng Li[2,5], Xiaoling Wei [2,5], Liuyang Sun [1,2], Zhiyuan Zhang[3], Chi Yang [3] ✉, Tiger H. Tao [1,2,4,5,6,7,8,9] ✉ & Zhitao Zhou [2,5] ✉

Taste, a pivotal sense modality, plays a fundamental role in discerning flavors and evaluating the potential harm of food, thereby contributing to human survival, physical and mental health. Patients with tongue cancer may experience a loss of taste following extensive surgical resection with flap reconstruction. Here, we designed a gustatory interface that enables the non-invasive detection of tongue electrical activities for a comprehensive operative assessment. Moreover, it decodes gustatory information from the reconstructed tongue without taste buds. Our gustatory interface facilitates the recording and analysis of electrical activities on the tongue, yielding an electrical mapping across the entire tongue surface, which delineates the safe margin for surgical management and assesses flap viability for postoperative structure monitoring and prompt intervention. Furthermore, the gustatory interface helps patients discern tastes with an accuracy of 97.8%. Our invention offers a promising approach to clinical assessment and management and holds potential for improving the quality of life for individuals with tongue cancer.

Humans navigate the world through five primary senses: vision, hearing, touch, smell, and taste. The taste buds, located on the tongue, play a crucial role in transducing chemical stimuli into nervous signals. These signals are subsequently conveyed to the gustatory center of the brain through peripheral neurons. The significance of the sense of taste in human evolution cannot be overstated, as it empowers individuals to recognize the five fundamental tastes – bitter, sweet, sour, salty, and umami. This ability not only facilitates the enjoyment of palatable foods but also aids in avoiding those with unpleasant tastes that may pose potential harm[1,2]. Beyond its evolutionary relevance, taste holds intricate connections with overall well-being. Taste disorders pose a threat to physical health, as an elevated taste threshold increases the susceptibility to cardiovascular diseases, diabetes, and other related health conditions[3,4]. Moreover, gustatory impairment extends its impact on mental health, leading to an elevated risk of depression or even suicide[5].

[1]2020 X-Lab, Shanghai Institute of Microsystem and Information Technology, Chinese Academy of Sciences, Shanghai 200050, China. [2]School of Graduate Study, University of Chinese Academy of Sciences, Beijing 100049, China. [3]Department of Oral Surgery, Shanghai Ninth People's Hospital, Shanghai Jiao Tong University School of Medicine; College of Stomatology, Shanghai Jiao Tong University; National Center for Stomatology; National Clinical Research Center for Oral Diseases; Shanghai Key Laboratory of Stomatology; Shanghai Research Institute of Stomatology; Research Unit of Oral and Maxillofacial Regenerative Medicine, Chinese Academy of Medical Sciences, Shanghai 200011, China. [4]Neuroxess Co. Ltd, Shanghai 200023, China. [5]State Key Laboratory of Transducer Technology, Shanghai Institute of Microsystem and Information Technology, Chinese Academy of Sciences, Shanghai 200050, China. [6]Center of Materials Science and Optoelectronics Engineering, University of Chinese Academy of Sciences, Beijing 100049, China. [7]Center for Excellence in Brain Science and Intelligence Technology, Chinese Academy of Sciences, Shanghai 200031, China. [8]Guangdong Institute of Intelligence Science and Technology, Hengqin, Zhuhai, Guangdong 519031, China. [9]Tianqiao and Chrissy Chen Institute for Translational Research, Shanghai, China. [10]These authors contributed equally: Xiner Wang, Guo Bai, Jizhi Liang, Qianyang Xie. ✉e-mail: yangchi63@hotmail.com; tiger@mail.sim.ac.cn; ztzhou@mail.sim.ac.cn

Tongue cancer ranks among the top 10 most prevalent malignancies globally, with an annual incidence exceeding 350,000 new cases[6]. The tumor can directly infiltrate the tongue tissue, leading to taste disturbances and significantly compromising the overall quality of life. In clinical settings, computed tomography (CT) emerges as the primary diagnostic tool, utilizing X-ray technology to generate images that facilitate the assessment of preoperative resection margins from tongue tissue conditions[7]. Despite its diagnostic utility, the increasing frequency of CT examinations raises concerns regarding the cumulative radiation dose and associated risks of hematological and lymphoid malignancies[8]. Following resection and reconstructive surgical procedures, the vigilant monitoring of the free flap assumes paramount importance in gauging the surgical outcome and promptly identifying situations necessitating intervention. Currently, flap observation predominantly relies on visual inspection, a method susceptible to subjective judgement and experiential biases. Simultaneously, studies have shown a deficiency of taste perception in the tongue regions subjected to reconstructive surgery with forearm flaps[9]. Notably, a greater extent of tongue resection is associated with increased damage to the glossopharyngeal nerve, resulting in diminished taste acuity and its detection threshold[10–12].

Previous studies show that prostheses have demonstrated promise in decoding and restoring various senses and sensorimotor functions[13], such as decoding visual representation from neural activities, achieving visual rehabilitation through artificial retinal implants[14–16], and enabling auditory sensations recognition and hearing recovery via cochlear implants[17,18]. With advancements in neuroengineering, speech and motor neuroprostheses have demonstrated their capability to decode cortical activity associated with attempted speech and voluntary motor impulses into text or command inputs for digital devices[19–22]. Despite these advancements, the field of prosthetics has yet to thoroughly explore gustatory decoding in patients with tongue cancer.

To address the unmet medical need, we designed a gustatory interface employing high-density, ultra-conformal tongue electrodes to capture electrical signals from the tongue and unravel physiological signals related to taste perception. The 256-channel electrodes offer complete coverage of the tongue surface, enabling high-resolution, non-invasive, and radiation-free recording, thereby introducing a groundbreaking approach for full-cycle tongue examination. The preoperative whole-tongue electrical mapping facilitates tumor localization and potential insights into a safe margin for surgical management. Postoperatively, ongoing tongue monitoring provides an objective assessment of tissue viability and fusion, ensuring meticulous evaluation of operation quality and allowing the clinician to pay close attention to the structure recovery of the reconstructed flap. As gustatory information is mapped onto the gustatory cortex on the surface of the brain, responses to different taste stimulations can be recorded by non-invasive electroencephalogram (EEG)[23]. Here, we present a multifunctional gustatory interface capable of high-performance taste decoding across patients with reconstructed tongue lack of taste buds (with an accuracy of 97.8%) through the integration of tongue electrophysiology and brain-computer interface (BCI). This interface offers significant potential to enhance operative assessment and facilitate the decoding of taste function for patients affected with tongue cancer.

## Results
### Overview of the multifunctional gustatory interface
The application scenarios of the multifunctional gustatory interface for patients with tongue cancer are described in Fig. 1, with flexible tongue electrodes serving as its most prominent component. The ultra-conformal, 256-channel electrodes adhere to the tongue surface, covering the tumor area to acquire tongue electrical (TE) signals evoked by taste stimulations. These signals, which capture the

responses of nerve fibers and taste receptors, exhibit exceptional spatial resolution and fidelity, rendering the gustatory interface invaluable throughout disease diagnosis, treatment, and recovery. Firstly, such bioelectronic device can create a spatial mapping of electrical activity across the tongue, enabling clinicians to localize the site of tongue cancer lesions and delineate the safe margin for clinical management (Fig. 1a). This introduces a paradigm for preoperative examination, promising to enhance surgical safety. Secondly, following resection and reconstructive surgery, postoperative structural monitoring can also be achieved through this interface, offering a more objective assessment compared to conventional subjective visual inspection (Fig. 1b). Notable differences in electrical signals from the tongue can be observed between the natural and reconstructed part, though these differences tend to decrease as tissues undergo growth and healing. While the structure of the reconstructed flap may gradually recover, the restoration of taste function remains unattainable, underscoring the importance of assisting patients in differentiating five tastes. Due to the gustatory system involving the tongue and its connection to the brain through the nervous system, the integration of tongue electrophysiology with BCI technologies stands as a robust approach for taste decoding for patients following irreversible tongue cancer surgical injury (Fig. 1c).

### Preoperative tongue cancer localization
The flexible tongue electrodes, a key component of the gustatory interface, are specifically designed and manufactured to ensure optimal biocompatibility, functionality, and superior conformability to the soft tongue surface (Fig. 2a and Supplementary Fig. 1). Theoretical simulations substantiate that electrodes with a thickness of 20 μm exhibit remarkable conformability to soft bio-tissues while maintaining sufficient mechanical strength for operation compared to other thicknesses (Fig. 2b and Supplementary Fig. 2). Furthermore, the low impedances of electrodes with a high yield and low-noise signal processing unit are critical for stable recording of TE signals with a high signal-to-noise ratio (SNR) (Fig. 2c and Supplementary Fig. 3). Another issue of crosstalk between the densely packed leads was investigated using finite element analysis (FEA) simulations and in vitro measurements. The results indicate that the parameters of our device effectively minimize crosstalk across various frequencies, thereby satisfying the requirements for electrophysiological recordings (Supplementary Figs. 4–6). The uniformity and stability of impedances and baselines for 256 recording sites between electrodes over time enable our tongue electrodes to form a conformal and reliable bioelectronic interface (Supplementary Figs. 7–10). The initial phase of developing the gustatory interface involved recording TE signals from normal subjects in various states to ascertain proper functionality (Supplementary Figs. 11, 12).

Preliminary experiments demonstrated that the gustatory interface is capable of producing an electrical mapping of the whole tongue through a non-invasive electrophysiological approach. This mapping serves as a valuable preoperative decision-making tool for patients with tongue cancer compared with radio imaging by CT[24]. Leveraging the outstanding adhesion to tissues and superior electrical performance of our 256-channel tongue electrodes, the interface is conformal on the entire surface of patients' tongue to capture taste-related electrical signals (Fig. 2d). Applying spectral analysis methods, which are now routinely used in electrophysiological studies, to the TE signals from the cancer site and its surrounding areas can facilitate the investigation of the intensity changes in time- and frequency-domain components related to the disease. Raw and effective time traces and time-frequency spectra from normal and cancer sites reveal fluctuation patterns of tongue electrical activities, with the cancer site displaying evident suppression in signal strength (Fig. 2e, f). The power spectral density (PSD) in Fig. 2g describes how the energy of the tongue electrical signals from the normal tissues and cancer site is

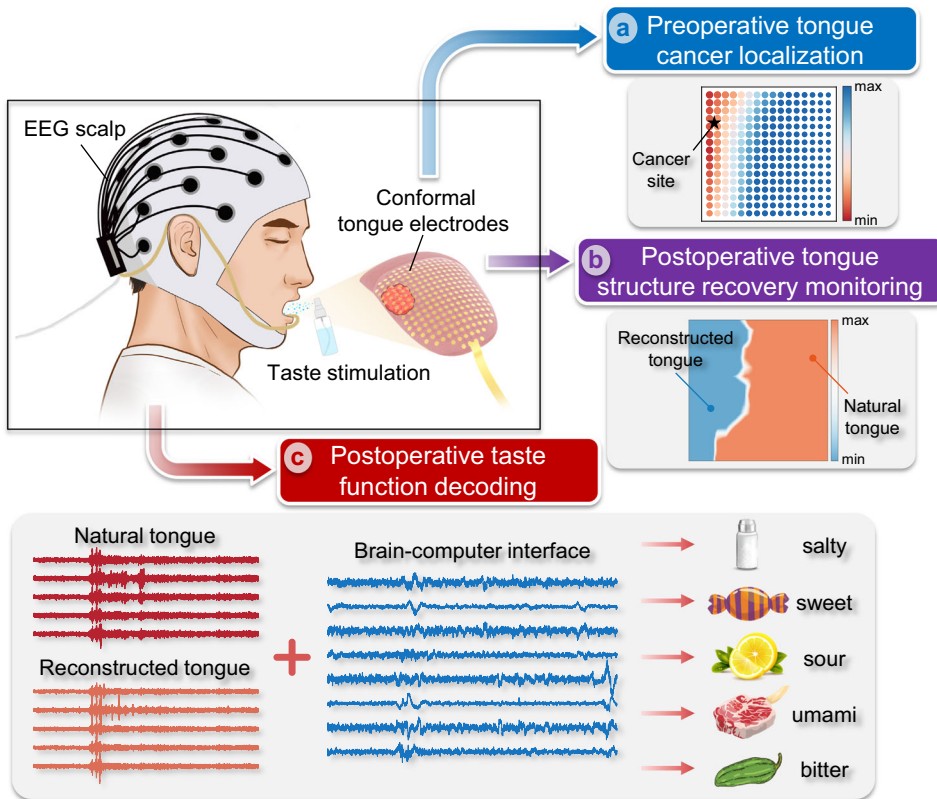

**Fig. 1 | Schematic overview of the gustatory interface.** Flexible and ultra-conformal tongue electrodes are placed on the tongue surface of a patient with tongue cancer to capture tongue electrical signals evoked by taste stimulations. These signals are subsequently employed in the following three scenarios:

**a** Preoperative tongue cancer localization. **b** Postoperative tongue structure recovery monitoring, and **c**, Postoperative taste function decoding with the assistance of brain-computer interface.

differently distributed with frequency. This phenomenon is attributed to cancer-mediated pathological changes and subsequent changes in electrical signals. By this means, the root mean square of amplitude has been calculated as the power and mapped into the actual position to visualize the cancer site localization (Fig. 2h), revealing a cancer central site with relatively weaker power, consistent with CT examination results in Fig. 2d. Similar findings were observed for the other four taste stimulations (Supplementary Fig. 13). Thus, through whole-tongue mapping, our gustatory interface effectively distinguishes between normal and cancerous tissues, providing critical information for clinicians to localize tumor sites and define the safe margins for surgical planning.

**Postoperative tongue structure recovery monitoring**

The primary treatment for patients diagnosed with tongue cancer involves surgical excision of the affected area, followed by reconstruction using free flaps. Postoperative monitoring conventionally relies on visual observation of the free flap's color and morphology, a method inherently subjective in nature. In contrast, our proposed interface offers an equally non-invasive method of monitoring the recovery of tongue structure through electrophysiological mapping. Specifically, the TE signals captured from both reconstructed and natural tongues, utilizing flexible tongue electrodes under different taste stimulations, are leveraged to assess the extent of recovery in the reconstructed part (Fig. 3a).

Following surgery, initial responses without taste stimulations reveal a discernible difference in the power spectral density of TE signals from reconstructed and natural tongues, particularly in the frequency range below 200 Hz (Fig. 3b). Notably, the reconstructed tongue exhibits slightly weaker amplitude and power compared to the

natural counterpart and similar results were observed in the case of taste stimulations (Supplementary Figs. 14–16). To illustrate the ability of the gustatory interface to monitor structure recovery over time, three patients were involved during two follow-up examinations after surgery, and we mapped the spatial distribution of TE signals evoked by taste stimulations through max-min normalization for each participant (Fig. 3c). Comparing the short-term and long-term recovery status post-surgery, normalized signal energy across all channels highlights the boundary between responses from the reconstructed and natural tongue parts. The uniform trend is that as the recovery time increases, the signal power on the reconstructed side increases, reducing the contrast between the two parts. Further quantitative analysis, computing the ratio between normalized power of the reconstructed and natural tongue, reinforces the results which imply a gradual approaching of TE activities in the reconstructed tongue towards those of the natural part while there exist individual differences in recovery between three patients (Fig. 3d). Therefore, our gustatory interface not only aids in preoperative assessment but also enables comprehensive monitoring through the postoperative period. The postoperative gustatory responses serve as critical prognostic indicators of tissue fusion and recovery and facilitate the evaluation of surgical quality.

**Taste function decoding from reconstructed tongue**

Upon completing structure rehabilitation, the subsequent crucial step is to decipher the impaired taste function. The human gustatory system extends beyond the tongue; its innervations, specifically the afferent nerve fibers, play a pivotal role in transmitting taste information to the gustatory cortex via synapses in the brainstem and thalamus[25–27]. To render the function of highly reliable and robust taste

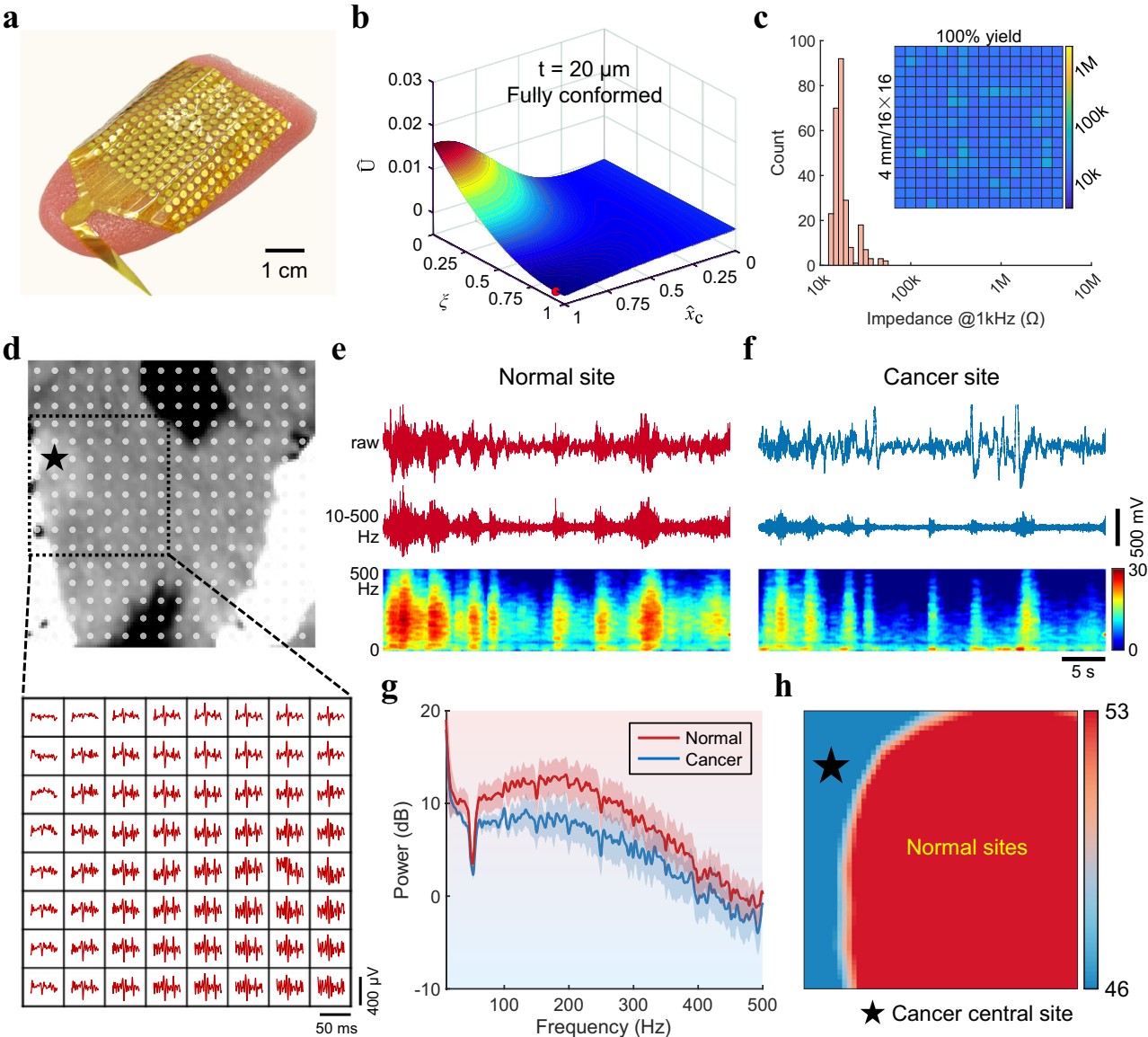

**Fig. 2 | Preoperative tongue cancer localization. a** Photograph of 256-channel tongue electrodes well-attached to the soft tongue surface. **b** Simulation of the normalized total energy of the system with the flexible tongue electrodes (20 μm thickness) laminated on the tongue. Global minima are highlighted with red dots. **c** Histogram of electrode impedance (at 1 kHz frequency) for the 256-channel tongue electrodes in (**a**). Inset: Spatial distribution of the impedance magnitude with respect to actual channel positions. **d** Preoperative CT image of the tongue from a patient with tongue cancer. The gray dots mark the positions of the 256

electrode channels. The waveform map of the cancer central site and a small area around it (black dashed box) is highlighted at the bottom. **e, f** Representative TE signals and time-frequency spectra from the normal site (**e**) and the cancer site (**f**). **g** Power spectral densities of TE signals from several channels in the cancer and normal sites in the 10–500 Hz range. Data are presented as mean values +/− SD. **h** Power heatmap of the black dashed box in (**d**), with the cancer central site indicated by the dark star.

decoding to the gustatory interface, we made full use of the close connection between the sensory organ and the cerebral nervous system. Accordingly, we simultaneously recorded EEG signals and tongue electrical activities, as depicted in the experimental paradigm detailed in Fig. 4a. Each trial started with a resting state followed by taste stimulations and synchronous recording. To mitigate any influence from previous trials, patients were instructed to rinse their mouths thoroughly and relax between two consecutive trials. A dual-modal fusion framework was designed for inferring gustatory information from EEG and TE signals, encompassing data preprocessing, feature extraction, discriminant correlation analysis (DCA) fusion, and final prediction (Fig. 4b). A critical component of the gustatory decoder is the modal fusion module, which integrates the features of EEG and tongue signals to generate a fused feature vector for predicting the perceived taste.

Based on the TE-EEG database of the gustatory responses from normal subjects (Supplementary Figs. 17–19), we optimized the model parameters by comparing different time-domain features and classifiers (Supplementary Figs. 20–23). The results in Supplementary Figs. 24–30 highlight the importance of integrating TE and EEG signals to achieve accurate taste classification. After validating the efficiency of the decoding framework, we applied the gustatory interface to patients with tongue cancer with the ultimate goal of decoding taste sensations from the reconstructed tongue. The recorded responses from the reconstructed tongue, along with EEG signals (Supplementary Fig. 31), were utilized to predict the present taste. Figure 4c visually illustrates the feature distribution after fusing TE data from the reconstructed tongue with EEG data to verify distinct taste-dependent states. We have examined various time- and frequency-domain

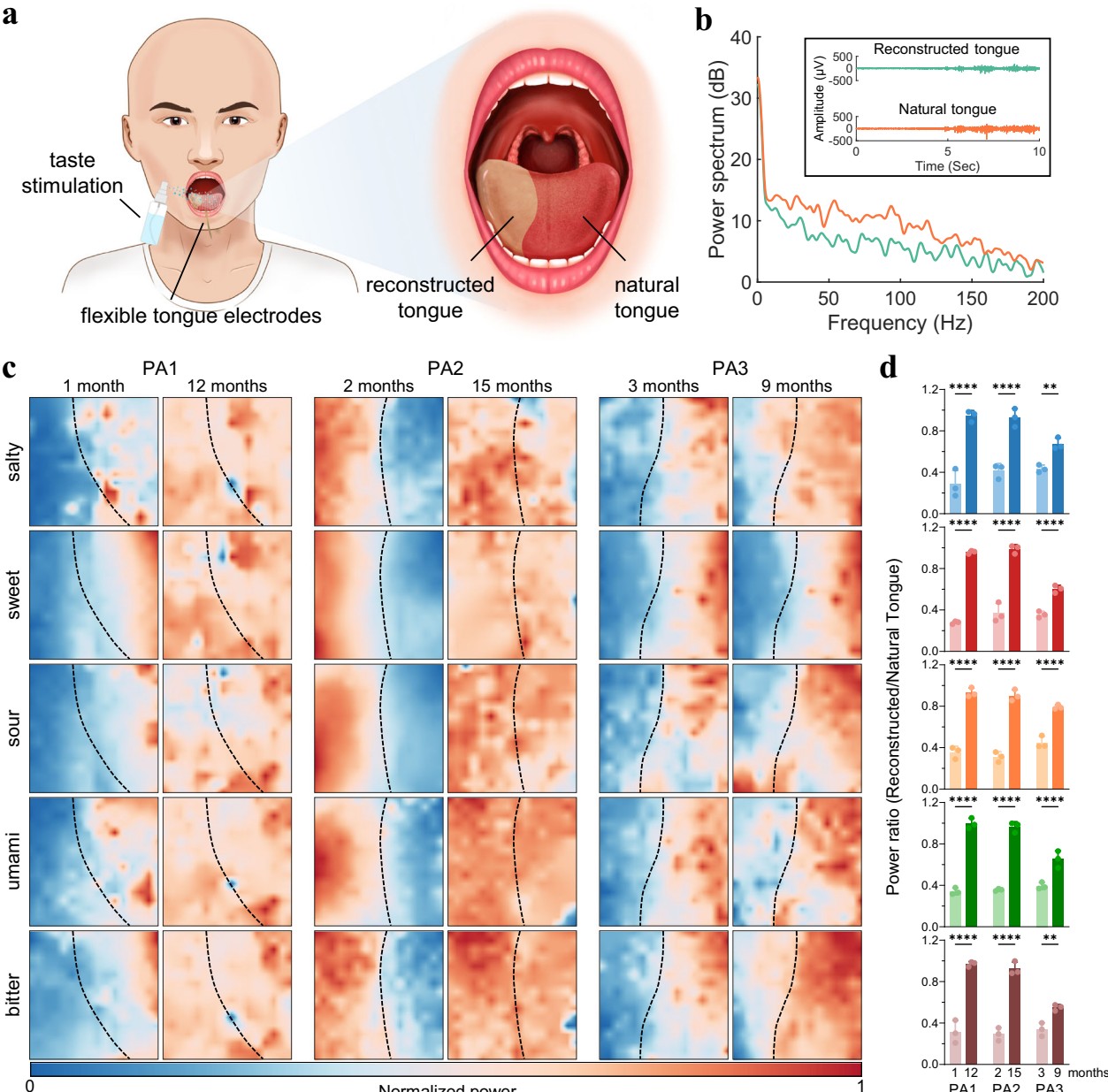

**Fig. 3 | Postoperative tongue structure recovery monitoring. a** Schematic illustration depicting the application scenario for structure recovery monitoring. The TE signals are acquired from the reconstructed and natural tongue of a patient who has undergone resection and reconstructive surgery. **b** Spectrum of TE signals from the natural and reconstructed tongue. The inset displays representative 10 s time traces from these two areas, with the first five seconds in a relaxed state and the second half in a tense state. **c** Normalized power heatmaps of taste-induced TE signals of three patients (PA1, PA2, and PA3) during two follow-up visits after surgery. The examinations were conducted at one and twelve months after surgery for PA1, at two and fifteen months after surgery for PA2, and at three and nine months after surgery for PA3. The boundary between the reconstructed and natural parts of the tongue is demarcated with dotted lines. For PA1 and PA3, the left side of the

dotted line refers to the reconstructed tongue while the right side of the line is the reconstructed tongue for PA2. **d** Bar plots illustrating the normalized power ratio between the reconstructed and the natural tongues under five taste stimulations for three patients during two follow-up examinations after surgery. Data are presented as mean values +/- SD. $p = 8.87 \times 10^{-7}$ for PA1, $p = 1.20 \times 10^{-5}$ for PA2, $p = 0.0090$ for PA3, salty; $p = 1.51 \times 10^{-9}$ for PA1, $p = 4.53 \times 10^{-9}$ for PA2, $p = 8.90 \times 10^{-5}$ for PA3, sweet; $p = 2.02 \times 10^{-8}$ for PA1, $p = 1.46 \times 10^{-8}$ for PA2, $p = 5.96 \times 10^{-6}$ for PA3, sour; $p = 3.10 \times 10^{-9}$ for PA1, $p = 7.41 \times 10^{-9}$ for PA2, $p = 7.40 \times 10^{-5}$ for PA3, umami; $p = 1.13 \times 10^{-7}$ for PA1, $p = 1.77 \times 10^{-7}$ for PA2, $p = 0.0057$ for PA3, bitter. (*$p < 0.05$, **$p < 0.01$, ***$p < 0.001$, ****$p < 0.0001$, two-way ANOVA).

features and several commonly used non-linear classifiers on the taste-evoked TE and EEG responses to assess the suitability of the model (Supplementary Tables 1, 2 and Supplementary Figs. 32, 33). The resulting confusion matrix in Fig. 4d exhibits a prominent diagonal, indicating excellent cross-patient decoding performance with an average accuracy of 97.8%. The dual-modal fusion framework

significantly improves accuracy compared to using only TE or EEG data (Supplementary Figs. 34, 35). These findings were also corroborated by individual patients (Supplementary Figs. 36–38). In summary, the interface introduces an innovative approach by accurately predicting gustatory information from electrophysiological signals of the reconstructed flap lacking taste buds, utilizing advanced dual-modal

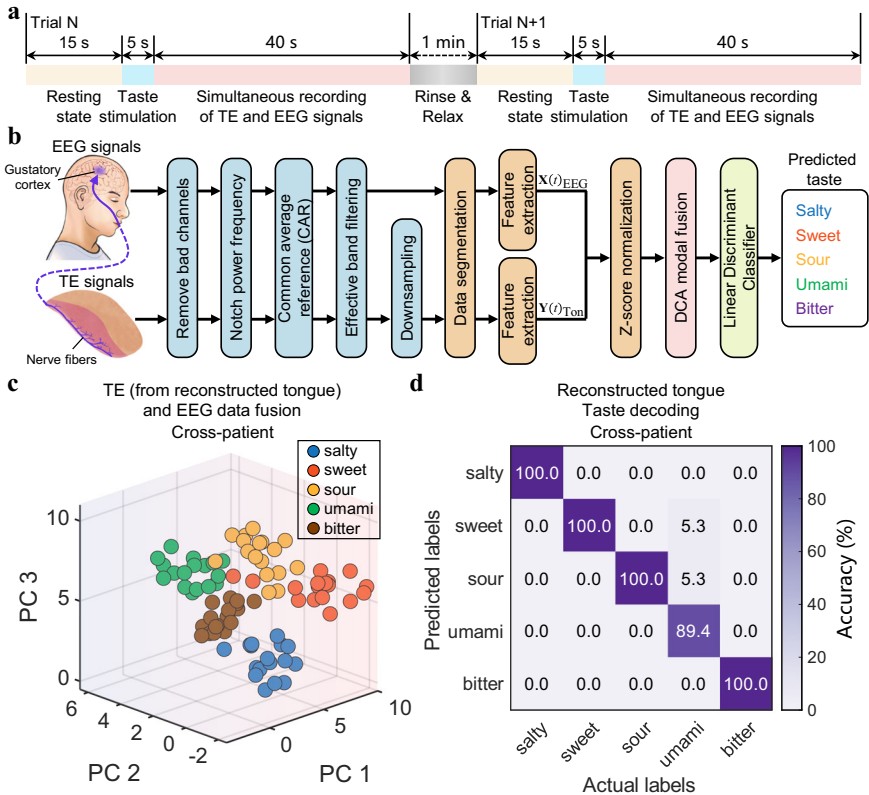

**Fig. 4 | Taste function decoding from the reconstructed tongue. a** Time progress bar illustrating the experimental paradigm. Each trial spans one minute, commencing with a 15-second resting state. Taste stimulation is applied between 15 and 20 s, followed by simultaneous recording of TE and EEG signals for 40 s. Patients were instructed to rinse and relax between two trials. **b** Schematic illustration of the TE and EEG signals processing pipeline and the dual-modal fusion algorithm for predicting five basic tastes. **c** Visualized result of the feature space after the fusion of TE data from the reconstructed tongue and EEG data. **d** Confusion matrix of cross-patient taste decoding (n = 5).

recording and decoding algorithms. This method holds the potential to assist patients in distinguishing flavors from unfamiliar and unknown foods.

## Discussion

The capacity to discern various tastes is a primary concern for patients with tongue cancer who have undergone extensive tumor resection and free flap reconstruction[28,29]. This study represents a pioneering effort in decoding taste function from the reconstructed tongue, employing a gustatory interface that deciphers tongue electrophysiological activities into taste-related information. This interface also establishes a framework for preoperative assessment and postoperative monitoring through non-invasive electrical mapping of the tongue.

Prior to surgery, our gustatory interface effectively localizes tongue cancer in alignment with CT results. The taste-induced electrical activity around the tumor expands the clinicians' understanding of tongue electrophysiological conditions beyond traditional imaging or ultrasound examinations[30]. This approach has the potential to unveil safe margins, thereby enhancing the safety of surgical resection. After the surgery, the whole-tongue electrical mapping derived from gustatory responses offers more reliable and objective insights into the viability and fusion of the reconstructed flap. This assists in evaluating surgical quality and monitoring structure recovery. Furthermore, our innovative strategy combines tongue electrophysiology with BCI to decode gustatory information from the reconstructed flap that lacks taste receptors.

The success here is powered by the conformal adhesion and excellent electrical performance of the flexible tongue electrodes.

These features enable the capture of TE signals evoked by chemical stimulations from the whole tongue with relatively high spatial resolution. Drawing inspiration from the human gustatory system, which includes the taste organ and gustatory cortex, we exploit the dual-modal fusion of TE activities and EEG signals, enabling precise taste function decoding.

Advances in tongue electrodes are expected to power higher-resolution recording, with the promise of more accurate and early diagnosis throughout the perioperative period. Meanwhile, the gustatory interface presents a promising avenue for establishing a database of tongue electrophysiology in the future, which can facilitate the extraction of comprehensive biomarkers, enabling the investigation of various diseases and their causes. In clinical settings, both the safety of the device and the patient's subjective experience are critical considerations. While our tongue electrodes may produce a degree of foreign body sensation, they are designed to be non-invasive and do not inflict pain or damage on patients. Although the device is still immature for clinical use, we would like to emphasize that it offers a proof-of-concept for clinical application related to the diagnosis and treatment of tongue cancer. Requirements for applications in clinical scenarios suggest future research directions, including the development of wireless bioelectronic interfaces. Another limitation of this study is that taste decoding currently involves a series of offline preprocessing and training, with results not immediately fed back to patients. Future endeavors will focus on establishing real-time taste decoding, reducing decoder training time, and employing more advanced models and algorithms to improve efficiency and performance. A closed-loop system with instantaneous feedback could achieve full capability for both recording and stimulation, enhancing

interaction with patients. Future gustatory prostheses that allow patients to perceive taste in real time would significantly contribute to improving patients' quality of life.

## Methods
### Participants
This research involved a total of seven patients in the Shanghai Ninth People's Hospital (5 males and 2 females). Specifically, three patients were diagnosed with tongue cancer who have not yet undergone surgery and five patients have undergone surgery of tongue resection and transplantation. One of the patients met the predetermined inclusion criteria both before and after the surgical intervention, who was diagnosed with primary tongue cancer and underwent subsequent tongue flap resection and reconstruction surgery. Three postoperative patients underwent two follow-up examinations to monitor the recovery using tongue electrical signals. The study received ethical approval from the Shanghai Ninth People's Hospital Institutional Review Board (IRB) of Shanghai Jiao Tong University School of Medicine (Approval No.:SH9H-2023-T174-2) and was conducted according to the guidelines of all relevant ethical regulations. In addition, three normal and healthy subjects (2 males and 1 female) were also involved in the task of taste classification to validate the feasibility of the system. No sex- or gender-based analysis is used in this study. The 256-channel high-density flexible tongue electrode arrays were attached to the tongue surfaces to record tongue electrical activities, while the participants wore an EEG cap for monitoring EEG signals. Prior to the experiment, all participants were informed that their involvement in the research was completely voluntary and that the task was conducted for research purposes. The procedural steps of the experiment were thoroughly explained, and each participant provided informed consent in written form.

### Fabrication of the tongue electrodes
The 256-channel electrode array is fabricated to form a sandwich structure with gold contacts embedded in flexible sheets of polyimide (PI) using a similar process in our previous work[31]. A 7 μm-thick substrate layer of PI (PI-2610, HD Microsystems, USA) was first spin-coated on the silicon wafer. After patterning a layer of photoresist (AZ 5214, AZ Electronic Materials USA Corp.), electron beam evaporation deposited a metal layer of 50 Å-thick chromium and 150 nm-thick gold. This was followed by a lift-off procedure in acetone that completed the pattern of recording sites and interconnectors. Another 13 μm-thick PI layer was spin-coated and cured, reserving as the encapsulation layer. Aluminum was sputtered as the etching mask and then patterned by UV photolithography, aluminum corrosion, and oxygen plasma dry etching to expose the openings areas and the perfusion holes, which perfuse tongue mucus and solution away from the electrode contacts. The fabrication process was completed by etching with the hydrofluoric acid to release the device from the silicon wafer. The final tongue electrodes have $16 \times 16$ recording sites with 1.3 mm in diameter and 0.9 mm in space. Bonding the flexible tongue electrodes to a flexible printed circuit board (FPC) could yield connection points for interfaces to our custom headstage, enabling simultaneous addressing and acquisition of 256 electrode channels.

### Mechanical simulation of the tongue electrodes
Conformability plays a significant role in the interface of the flexible electrodes laminated on the soft tongue surface in that improved conformability can lead to higher adequate interfacial adhesion strength and higher signal-to-noise ratio. To simplify the analysis, the interface is modeled as a thin elastic membrane laminated on a sinusoidally corrugated substrate. The conformability $\hat{x}_c$ is defined as the ratio between the horizontal projection of the contact zone and half of the substrate wavelength, which can be derived through the energy

minimization principle. The total energy is given as

$$U_{\text{total}} = U_{\text{bending}} + U_{\text{membrane}} + U_{\text{adhesion}} + U_{\text{substrate}} \tag{1}$$

where $U_{\text{bending}}$, $U_{\text{membrane}}$, $U_{\text{adhesion}}$ and $U_{\text{substrate}}$ denote the bending energy of the membrane, the membrane energy related with tensile strain, the interfacial adhesion energy and the elastic energy stored in the substrate, respectively.

Through analytical and dimensionless computation, the final normalized total energy can be derived as a function of $\hat{x}_c$ and the degree of deformation of the tissue $(\xi)$[32,33].

$$\hat{U} = \alpha \frac{\xi^2}{12} \eta^3 D(\hat{x}_c) + \alpha \eta \xi^4 \beta^2 K(\hat{x}_c, \xi\beta) - \frac{\mu}{\beta^2} E(\hat{x}_c, \xi\beta) + \frac{(1-\xi)^2}{4\pi} \left[ F_1(\hat{x}_c) - F_2(\hat{x}_c) \right] \tag{2}$$

where $\alpha$, $\mu$, $\eta$ and $\beta$ are four input parameters, which denote the modulus ratio between the membrane and the substrate, the normalized membrane–substrate intrinsic work of adhesion, the normalized membrane thickness and the normalized roughness of the substrate, respectively. $\xi$ is the ratio between the initial amplitude and the deformed one. When the four input parameters are given, the total energy with respect to $\hat{x}_c$ and $\xi$ can be visualized as a three-dimensional landscape plot. The value of $\hat{x}_c$ corresponding to the global minimum can indicate the degree of conformability. $\hat{x}_c = 0$ denotes the non-conformed state, while $\hat{x}_c = 1$ denotes the fully conformed state. The in-between value suggests partial conformability. As the thickness of the device rises, the result goes from complete conformability to partial conformability.

### Electrical characterization of the tongue electrodes
The crosstalk simulation was conducted using the COMSOL Multiphysics 6.0 FEA software. A 2D simulation model was built with customized mesh sizes, including two electrode traces, one substrate layer, one insulator layer, and a large circular ambient. An input voltage of 1 V was applied to one electrode trace, and the other was grounded.

Electrochemical impedance spectroscopy (EIS) measurements were performed on a CHI660e electrochemical workstation (CHI660e, CH Instruments Inc., China). The test was conducted in $1 \times$ PBS (phosphate-buffered saline) with pH 7.4 at room temperature, using a three-electrode configuration with an Ag/AgCl electrode as the reference electrode and a Pt wire as the counter electrode.

### Taste stimulation and delivery
To simulate taste experiences that closely resemble those encountered in daily life and with a comprehensive reference to other studies[34–38], the aqueous solution of condiments and food-grade additives was used with a clear state and the same concentration of 0.5 M to ensure consistency in the experimental variables, including sodium chloride (salty, Morton), sucrose (sweet, Taikoo), citric acid (sour, Tanggui), sodium glutamate (umami, Youbaojia) and magnesium chloride (bitter, Tanggui). Patients were given the option to select between quinine and $MgCl_2$ as bitter stimuli during their enrollment in the clinical trials, with detailed information provided for both solutions. All patients opted for $MgCl_2$ as the testing reagent. Taste stimuli solutions with different concentrations were investigated for taste classification, which suggests that the decoding performance is not affected by the concentration (Supplementary Fig. 39). Taste stimulations were delivered with a spray bottle to apply two pumps of solution in each trial. The position of the nozzle can be freely adjusted to ensure that the spray covers the targeted area of the tongue surface.

### Experimental procedure
Each trial started with a fifteen-second epoch where the participants were instructed to make the tongue and the oral cavity in a completely

resting state. Then the taste stimulations were applied on the tongue for the next five seconds. After the stimulation, the participants were instructed to back the tongue and keep it still. During the recording process, the participants were comfortably seated and remained still to exclude the interference of motion artifacts. The experiment for each participant contained three or five sessions depending on the physiological state of the volunteers, and each session included five trials corresponding to five basic tastes. The participants gargled with water and had a short break between different taste stimulations to eliminate the crosstalk between different tastes.

### Electrophysiological recording
The TE signals from our flexible electrodes were amplified and digitized using a custom two-layer stacked 256-channel headstage, which was connected to a multichannel data acquisition system CereCube NSP8 (Neuroxess Co., Ltd., China). The reference and ground electrodes were placed on the bilateral faces of the participants with adhesive electrodes. During the data collection procedure, the TE signals were recorded at a sampling rate of 4000 Hz.

The EEG data were recorded using a gel-free electrode cap connected to the OpenBCI Cyton Biosensing Board (OpenBCI, NY, USA). The signals were recorded from 6 channels (C4, P3, P4, T3, T4, T5), whose positions are according to the international 10-10 international system. The ground and reference electrodes were located in the position of AFz and CPz, respectively. The sampling frequency of EEG signals was 250 Hz.

### Data processing and feature extraction
The TE and EEG signals were first visually and quantitatively inspected for mechanical artifacts and excess noise to remove bad channels. Notch filters were adopted to remove the power frequency of 50 Hz and its possible harmonics. A standard technique, common average reference (CAR), was used to reduce shared common-mode noise in multi-channel data[39]. Since bioelectrical artefacts typically occur in higher frequency ranges, while mechanical motion artefacts generally exhibit frequency spectra below 10 Hz[40], we have employed algorithmic post-processing interventions to separate the contributions of these overlapping components based on their characteristic frequencies[41]. Specifically, our study utilized finite impulse response (FIR) filters with upper and lower cutoff frequencies of 10 Hz and 500 Hz to minimize the influence of both bioelectrical and mechanical artefacts, thereby capturing valuable electrophysiological signals relevant to gustatory stimuli. EEG signals were bandpass-filtered with a range of 1–40 Hz. Since the sampling rates of the two signals differ, the TE signals were down-sampled for event alignment. To exclude the interference from hand and mouth movements during taste stimulation, the processed TE and EEG data streams were truncated with segments of 20–50 s for the purpose of feature extraction and further analysis.

Feature extraction is one of the essential steps in realizing pattern recognition. After a comprehensive comparative analysis with frequency-domain features and their combinations, time-domain features are preferred in this study due to their computational simplicity and ease of implementation without any need for data transformations. The taste-induced features of the TE and EEG data were extracted for the next steps of dual-modal fusion and classification. Herein, the following five time-domain features are given as

$$RMS = \sqrt{\frac{\sum_{i=1}^{L} x_i^2}{L}} \tag{3}$$

$$MAV = \frac{1}{L}\sum_{i=1}^{L}|x_i| \tag{4}$$

$$Var = \frac{1}{L}\sum_{i=1}^{L}(x_i - \mu)^2 \tag{5}$$

$$Sk = \frac{1}{L}\sum_{i=1}^{L}\left(\frac{x_i - \mu}{\sigma}\right)^3 \tag{6}$$

$$Ku = \frac{1}{L}\sum_{i=1}^{L}\left(\frac{x_i - \mu}{\sigma}\right)^4 \tag{7}$$

where $x_i$ denotes the $i^{th}$ data point in each window, $L$ denotes the length of the sliding window, $\mu$ and $\sigma$ are denoted as the mean value and standard deviation of all the data points in a sliding window.

$\mathbf{X}(t)_{EEG}$ and $\mathbf{Y}(t)_{Ton}$ are extracted from processed signals using the sliding window method with lengths of 0.2 s, 0.4 s, 0.6 s, 0.8 s and 1 s and normalized using the Z-score method across different recording sessions.

### Dual-modal fusion algorithm
Modal fusion is a critical step in a hybrid biometric system. In this study, the discriminant correlation analysis (DCA) method was adopted to transform the two feature vectors that incorporate relevant information associated with the TE and EEG activities into a single feature vector, which is anticipated to enhance the accuracy of prediction[42,43]. Two matrices $\mathbf{X}(t)_{EEG}$ and $\mathbf{Y}(t)_{Ton}$ were extracted from the EEG and TE signals with the dimension of $p \times n$ and $q \times n$, where $p$ and $q$ corresponded to the feature dimension of these two feature sets respectively. Firstly, the classes within each set were separated by utilizing the between-class scatter matrix of a single modality. Secondly, pair-wise correlations across these two feature sets were maximized by diagonalizing the between-set covariance matrix using the singular value decomposition (SVD) method. Thus, the two transformed matrices were obtained and the final feature set was the summation of the two transformed ones.

### Taste decoding strategy from the reconstructed tongue
For each patient after surgery, the transformed features after fusing the TE signals from the natural tongue and EEG data constitute the training dataset due to the sensitivity of the natural tongue to tastes, while the testing set is composed of the TE signals from the reconstructed tongue and EEG data after DCA fusion. After comparing with other non-linear classifiers, the multiclass linear discrimination analysis (LDA) classifier was adopted to identify and predict which taste is applied. The classifier was also applied to the dataset across five patients to validate the generalizability and scalability of the algorithm. The LDA classifier was finally selected due to its relatively optimal performance. It is also more likely to achieve real-time applications by virtue of the low complexity in the future.

### Statistical analysis
The experiments involving statistical tests were conducted with a minimum of $N = 3$ trials. We used two-way ANOVA to analyze whether the power ratios between the TE signals of the reconstructed tongue and natural tongue during two postoperative examinations for three patients have significant differences.

### Reporting summary
Further information on research design is available in the Nature Portfolio Reporting Summary linked to this article.

## Data availability
All data supporting the findings of this study are available within the article and its supplementary files. Any additional requests for

information can be directed to, and will be fulfilled by, the corresponding authors. Source data are provided with this paper.

## Code availability

The custom MATLAB scripts used for electrophysiology analysis are available from Zenodo (https://doi.org/10.5281/zenodo.13895193).

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

## Acknowledgements

This work was partially supported by the National Science and Technology Major Project from the Minister of Science and Technology of China (Grant No. 2018AAA0103100, T.H.T.), Youth Innovation Promotion Association for Excellent Members, CAS (Grant No. Y2023070, Z.Z.), Shanghai Rising-Star Program (Grant Nos. 22QA1410900, Z.Z.; 23QC140100, G.B.), National Natural Science Foundation of China (Grant No. 82271038, G.B.), Key Research Program of Frontier Sciences, CAS (Grant No. ZDBS-LY-JSC024, T.H.T.), Shanghai Pilot Program for Basic Research-Chinese Academy of Science, Shanghai Branch (Grant No. JCYJ-SHFY-2022-01, T.H.T.), Shanghai Municipal Science and Technology Major Project (Grant No. 2021SHZDZX, L.S.), CAMS Innovation Fund for Medical Sciences (Grant No. 2019-I2M-5-037, Z.Y.Z.), Ministry of Science and Technology of the People's Republic of China (Grant No. 2023YFC2509100, C.Y.), the Special Project for Clinical Research of Shanghai Municipal Health Commission (Grant No. 20204Y0459, Q.X.), the Innovative Research Team of High-level Local Universities in Shanghai, the Jiangxi Province 03 Special Project and 5 G Project (Grant No. 20212ABC03W07, Z.Z.).

## Author contributions

X.W., G.B., J.L., and Q.X. contributed equally to this work. Z.Z., T.H.T., and X.W. conceived the idea. Z.Z., T.H.T., C.Y., Z.Y.Z., and X.W. designed the experiments. Z.C. fabricated the tongue electrodes. X.W., G.B., J.L., Q.X., Z.Z., L.S., and X.L.W. performed the experiments. X.W., Z.Z., G.B., E.Z., and M.L. analyzed the data. X.W., Z.Z., and T.H.T. wrote the paper. All authors discussed the results and provided comments for the manuscript.

## Competing interests

The authors declare no competing interests.
