## [Peer Review File · Nature Communications]

REVIEWER COMMENTS

Reviewer #1 (Remarks to the Author):

This manuscript presents a gustatory prosthesis technology for reconstructing taste function in patients post tongue cancer surgery. By integrating high-density flexible tongue electrodes with brain-computer interface (BCI) technology, the study successfully recorded and analyzed tongue electrophysiological activities and EEG signals to identify and predict taste stimuli. The research utilized time-domain feature extraction, discriminant correlation analysis (DCA) for modal fusion, and employed a multiclass linear discriminant analysis (LDA) as the classifier. The application of this technology in patients with tongue cancer demonstrated a high accuracy in taste function assessment, offering robust data support for preoperative and postoperative tongue structure monitoring. I have several concerns regarding certain research strategies and specific technical approaches of the manuscript, as follows.

1. Regarding the detection and assessment of TE signals, is it possible to differentiate between taste dysfunction caused by tongue cancer and that caused by other factors such as inflammation or medication?
2. What is the biocompatibility of the tongue electrodes? Has there been an assessment of harmful residual following processes such as photolithography and other micromanufacturing techniques, along with corresponding biocompatibility treatments? Were necessary animal experiments or third-party evaluations conducted to assess the biocompatibility of the tongue electrodes before human trials?
3. The manuscript frequently mentions postoperative taste reconstruction, but the current research is limited to post-surgical sensing and decoding of TE signals (combined with EEG signals) without conducting clinical studies on active taste reconstruction based on postoperative taste assessments, such as feedback adjustment, stimulation, or pharmacological interventions. Therefore, consider whether the term "taste reconstruction" is appropriate.
4. In experiments involving taste stimulation of subjects, detailed TE and EEG signals were recorded to construct a taste model. Should the subjective experiences of the subjects also be considered as part of the baseline or reference for the taste model, to better accommodate individual differences in taste decoding?
5. The study involves three preoperative patients and five postoperative patients. Were there any participants who took part in the clinical trials both before and after surgery? If so, please specify. If not, how does the constructed model account for individual differences among patients?
6. How uniform are the TE signals measured by each electrode in the tongue electrode array? Has there been an effort to measure and establish a sensing baseline for each electrode? Furthermore, the manuscript uses normalized data to plot heat maps, such as Figure 3d. During normalization, is a unified baseline used, or is an independent baseline for each electrode applied, or perhaps a baseline derived from averaged processing?
7. In the synchronous measurement of the TE array, how is signal crosstalk between adjacent electrodes

handled?

8. As described by the authors, the collection of TE signals needs to be carried out multiple times both before and after surgery. Have the authors considered the tongue electrodes to be more durable "instruments" or easily replaceable "consumables"? If they are the former, how is the physical, chemical state, and performance stability of the tongue electrodes over time, and what is the data repeatability between multiple measurements? If they are the latter, how do you ensure the uniformity of signal measurements between electrodes prepared in the same or different batches?

9. It is mentioned that 0.5 M solutions of five different tastes were used as taste stimuli. However, based on the typical ranges for daily diets, as well as the literature cited in references 30 and 31, the concentration of 0.5 M for citric acid and magnesium chloride to provide sour and bitter stimuli appears to be excessively high. It may even cause discomfort for the subjects. Similarly, the concentration of 0.5 M glutamate sodium for umami taste is also relatively high. Could the authors explain the rationale behind using such high concentrations of taste stimuli solutions and its impact on the experimental results?

10. The authors have opted for time-domain features due to their computational ease. While this choice is justified for initial studies, the complexity of biological signals might be better represented using a combination of both time and frequency-domain features. It would be beneficial to discuss the potential advantages or results of incorporating such features and whether any preliminary investigations in this direction were conducted.

11. Linear Discriminant Analysis (LDA) is chosen for classifying taste signals. Although LDA is known for its simplicity and efficiency, it might not be the most robust option for complex signal classification tasks. Consideration of non-linear classifiers, such as Support Vector Machines, Random Forests, or even deep learning models, could potentially enhance classification performance. A comparative study of these models against LDA, with relevant performance metrics, would provide a more comprehensive understanding of the classifier's suitability.

Reviewer #2 (Remarks to the Author):

The authors purport to have developed a novel device for detecting and directing surgical management of lingual cancer. This device is a thin, flexible, 256 electrode array that is applied to the surface of the tongue and that is designed to record taste-evoked responses from the lingual surface. In addition to being designed to assess lingual cancer, the authors claim that the device has the potential to enhance taste by combining their technology with EEG recordings (brain computer interface, BCI).

Unfortunately, the report has several flaws that lead one to question the significance and relevance of the findings. First, the authors claim to have recorded taste-evoked responses from the electrode array (Fig 3c-e). The figure appears to show data from one patient, 3 and 9 months after surgical removal of lingual cancer, although the details are scanty. Thus, the author's statement about Statistical Analysis was unconvincing ("All experiments were conducted with a minimum of N = 3 for each data point").

Moreover, the taste stimuli are highly unusual and inappropriate. Namely, for testing taste-evoked responses, Wang et al applied solutions of 500 mM NaCl (salty), 500 mM sucrose (sweet), 500 mM citric acid (sour), 500 mM sodium glutamate (umami) or 500 mM magnesium chloride (bitter). The authors claim these stimuli and these concentrations those used in other studies (refs 30, 31). However, only NaCl and sucrose stimuli were similar to those published in refs 30, 31. Contrary to the authors' claim, Wang et al applied much higher concentrations of taste stimuli than used in the publications they cited (e.g. the cited publications elicited sour with 39 mM citric acid and bitter with 0.2 mM quinine), or not tested at all (sodium glutamate).

Second, and more importantly, the electrical activity of taste buds and/or taste afferents (Fig 2C) were unconvincing. There is no marking for when the taste stimuli were applied and no way to differentiate electrical signals from other sources (thermal, tactile, or more likely, EMG). Moreover, the topography of the electrical signals (Fig.2D) doesn't appear to reflect any particular distribution of taste buds.

Third, Wang et al did not explain the significance of "power" and "frequency spectrum" in the recordings very well (Fig 2b,e, f, g,h; Fig 3b,c,d,e). If these are significant measurements, the authors should describe and discuss what the data signify.

Fourth, the potential use for such a device in a clinical environment is dubious. Oral cancer patients experience significant mechanical allodynia. Applying and using a prosthetic device such as Wang described, regardless of how soft and pliable is the array, is likely to be quite painful and rejected. The authors did not discuss this aspect of their device.

Point by point response (comments in black and responses in blue):

Reviewer #1:

[1] This manuscript presents a gustatory prosthesis technology for reconstructing taste function in patients post tongue cancer surgery. By integrating high-density flexible tongue electrodes with brain-computer interface (BCI) technology, the study successfully recorded and analyzed tongue electrophysiological activities and EEG signals to identify and predict taste stimuli. The research utilized time-domain feature extraction, discriminant correlation analysis (DCA) for modal fusion, and employed a multiclass linear discriminant analysis (LDA) as the classifier. The application of this technology in patients with tongue cancer demonstrated a high accuracy in taste function assessment, offering robust data support for preoperative and postoperative tongue structure monitoring.

We sincerely thank the reviewer for the positive comments.

[2] I have several concerns regarding certain research strategies and specific technical approaches of the manuscript, as follows.

Regarding the detection and assessment of TE signals, is it possible to differentiate between taste dysfunction caused by tongue cancer and that caused by other factors such as inflammation or medication?

We thank the reviewer for the insightful comment. We would like to clarify that during the experimental design and patient selection process, careful consideration has been given to the inclusion of individuals diagnosed with primary tongue cancer while deliberately excluding those who had undergone prior pharmacological interventions. Rigorous measures have been implemented to mitigate confounding variables that might interfere with the factors causing taste dysfunctions.

Additionally, disease-related features can potentially be interpreted from TE signals by utilizing electrophysiological analysis methods from disciplines such as neuroscience. An exploratory experiment has been conducted to investigate the tongue electrophysiology related with the taste disorders caused by some special factors, such as inflammation. We have recorded tongue electrical activities of one subject with taste dysfunction due to COVID-19, with the inflammatory response leading to cellular and genetic changes that could alter taste as a possible mechanism [Lozada-Nur, F. et al. *Or. Surg. Or. Med. Or. Pa.* 130(3), 344-346 (2020)]. We first compared the spatial distribution of the TE signals captured from one patient with tongue cancer (**Figure R1Error! Reference source not found.a**) and one subject infected with COVID-19 (**Figure R1Error! Reference source not found.b**) at the same scale. It can be found that there is a clear and regular demarcation between the tongue cancer central site and the normal tissues in term of the TE signal intensities, while the electrical activities exhibit an irregular distribution throughout the whole tongue of the infected COVID-19 subject who suffered from taste loss. **Figure R1Error! Reference source not found.c** illustrates the power distribution of TE signals from the cancer central site and putative inflamed site, which indicates a difference in the frequency-domain characterization. Thus, the analysis of tongue electrophysiology provides a valuable tool for differentiating between taste dysfunction caused by tongue cancer and other factors such as inflammation. The above findings offer a potential pathway for establishing a database of tongue electrophysiology in the future, from which we

can extract more comprehensive biomarkers to distinguish different diseases and causes.

Figure R1. Analysis of the tongue electrical signals from subjects with different diseases. The spatial distribution of TE signals captured from one patient with tongue cancer (a) and one subject infected with COVID-19 (b). c, Power spectral densities of the signals from the cancer central site and the putative inflamed site related with COVID-19.

[3] What is the biocompatibility of the tongue electrodes? Has there been an assessment of harmful residual following processes such as photolithography and other micromanufacturing techniques, along with corresponding biocompatibility treatments? Were necessary animal experiments or third-party evaluations conducted to assess the biocompatibility of the tongue electrodes before human trials?

We thank the reviewer for the valuable comments. During the fabrication process, the tongue electrodes underwent multiple cleaning cycles using flowing deionized water (DI water) to minimize the presence of residual contaminants. Upon completion of the electrodes fabrication, the devices were sterilized by Co-60 gamma-ray irradiation with doses of 25 kilo Gray (kGy) before used on humans.

We commissioned a third-party testing organization to quantify the residue of additives present on the electrodes. The methodology encompassed immersing the electrode in ultrapure water maintained at a temperature of 37°C, equivalent to human body temperature, for a period of 24 hours. Following this immersion, an exhaustive analysis of the leach solution was undertaken to identify its constituents. The results revealed the absence of hazardous compounds used in the electrode preparation process, with the findings presented in **Figure R2**.

Following the fabrication within an ultra-clean room environment, the assembly of tongue electrodes was conducted in a 100,000-level Good Manufacturing Practice (GMP) purification workshop for the production of implantable medical devices. Subsequently, we engaged the Jiangsu Medical Device Inspection Institute, a certified third-party institution, to conduct a comprehensive suite of biocompatibility tests on the device. These tests encompassed cytotoxicity, intradermal response, skin sensitization, pyrogenicity, acute systemic toxicity, hemolysis, among others, in accordance with the *National Standard of the People's Republic of China GB/T 16886 Biological evaluation of medical devices* (specifically relating to *Part 4: Selection of tests for interactions with blood, Part 5: Tests for in vitro cytotoxicity, Part 10: Tests for irritation and skin sensitization, Part 11: Tests for systemic toxicity*). The tongue electrodes successfully met the criteria for all tests administered (**Figure R3**).

Furthermore, we conducted comprehensive efficacy and safety assessments of flexible electrodes on large animal models at the esteemed third-party institution, WuXi AppTec. These

electrodes were surgically implanted into the cerebral cortex of Labrador dogs, which puts forward higher requirements for biocompatibility compared to non-invasive tongue electrophysiology. The evaluations have been successfully concluded, and the results are delineated in **Figure R4**.

Following a comprehensive review of the aforementioned test reports pertaining to the biological safety and efficacy of the electrodes, we have secured the requisite ethical approval to initiate the clinical trial involving tongue cancer patients at Shanghai Ninth People's Hospital, Shanghai Jiao Tong University School of Medicine. The ethics committee approval letter is shown in **Figure R5**, which we have mentioned in the Methods of the original manuscript.

Figure R2. Analysis report of the residual components in the leach solution of the electrodes

江苏省医疗器械检验所
检验报告

报告编号: 2021QW1435 S 共1页 第1页

序号	检验项目	条款	要求	检验结果	单项结论	备注
1	细胞毒性 Cytotoxicity	/	取样品的电极部位,按 6cm ² /mL(表面积为电极两面表面积之和)比例,在 (37±1) °C, (24±2) h 浸提条件下制备试验液,浸提介质: 含血清 MEM 培养液,取试验液按照 GB/T16886.5-2017 中规定的方法进行试验,细胞存活率应不小于 70%。	102.8%	符合 PASS	/
2	皮内反应 Intradermal reaction	/	取样品的电极部位,按 6cm ² /mL(表面积为电极两面表面积之和)比例,在 (50±2) °C, (72±2) h 浸提条件下制备试验液,浸提介质: 0.9%氯化钠注射液,浸提介质: 0.9%氯化钠注射液;精制棉籽油,取试验液按照 GB/T16886.10-2017 中规定的方法进行试验,试验样品最终得分不大于 1.0。	0.9%氯化钠注射液 浸提液组: 0 棉籽油浸提液组: 0	符合 PASS	/
3	皮肤致敏 Skin sensitization	/	取样品的电极部位,按 6cm ² /mL(表面积为电极两面表面积之和)比例,在 (50±2) °C, (72±2) h 浸提条件下制备试验液,浸提介质: 0.9%氯化钠注射液;精制棉籽油,取试验液按照 GB/T16886.10-2017 中规定的豚鼠最大剂量试验方法进行试验,应无皮肤致敏反应。	0.9%氯化钠注射液 浸提液组: 无皮肤致敏反应 棉籽油浸提液组: 无皮肤致敏反应	符合 PASS	/
4	热原 Pyrogenicity	/	取样品的电极部位,按 6cm ² /mL(表面积为电极两面表面积之和)比例,在 (50±2) °C, (72±2) h 浸提条件下制备试验液,浸提介质: 0.9%氯化钠注射液,取试验液按照《中华人民共和国药典》2020 版四部 1142 热原检查法进行试验,热原检查应无热原反应。	热原检查符合规定	符合 PASS	/
5	急性全身毒性 Acute systemic toxicity	/	取样品的电极部位,按 6cm ² /mL(表面积为电极两面表面积之和)比例,在 (50±2) °C, (72±2) h 浸提条件下制备试验液,浸提介质: 0.9%氯化钠注射液;精制棉籽油,取试验液按照 GB/T16886.11-2011 中规定的方法进行试验,应无急性全身毒性反应。	0.9%氯化钠注射液 浸提液组: 无急性全身毒性反应 棉籽油浸提液组: 无急性全身毒性反应	符合 PASS	/

以下空白

检验: 廖峰 刘玉 马洁 审核: 沈芳

江苏省医疗器械检验所
检验报告

报告编号: 2021QW3271 S 共1页 第1页

序号	检验项目	条款	要求	检验结果	单项结论	备注
1	溶血 Hemolysis	/	取样品的电极部位,按 6cm ² /mL(表面积为电极两面表面积之和)比例制备供试品,按照 GB/T16886.4-2003 规定的试验方法进行试验,供试品溶血率应小于 5%。	0.9%	符合 PASS	/

以下空白

检验: 廖峰 审核: 沈芳

Figure R3. Test reports of our tongue electrodes for cytotoxicity, intradermal reaction, skin sensitization, pyrogenicity, acute systemic toxicity, and hemolysis.

FINAL REPORT

DISPOSABLE ELECTROCORTICOGRAPHY (ECOG) ELECTRODE: A FEASIBILITY, SAFETY AND EFFECTIVENESS STUDY IN LOCALIZING FUNCTIONAL BRAIN AREAS OF LABRADOR CANINE

STUDY NO. E59-0001-SGC

TEST CODE LC200003.1

TESTING FACILITY WUXI APPTec (SUZHOU) CO., LTD MEDICAL DEVICE TESTING CENTER NO.818 WUSONG ROAD, WUZHONG DISTRICT, SUZHOU 215122, JIANGSU, CHINA

SPONSOR JIANGXI NAOHU TECHNOLOGY CO., LTD. FLOOR 1, BUILDING 1-2#, PHASE II, ENTREPRENEURSHIP INCUBATION BASE FOR MEDIUM, SMALL AND MICRO ENTERPRISES NANCHANG, JIANGXI, CHINA, 330029

FINAL REPORT DATE 2022-01-28

surgery, and residual bone dregs were observed in the bleeding area, considering the mechanical system injury caused by the bone dregs. Therefore, none of the above pathological changes were considered to be related to the test articles.

In summary, the histopathological changes associated with the test articles were minimal to mild neutrophil infiltration of the dura mater, leptomeninges, and arachnoid membrane immediately after surgery; the histopathological changes observed in the blank and test article group were similar after 4 weeks of surgery, and the histopathological changes associated with the test articles were not observed. No pathological changes were found in the brain tissue of animals immediately after surgery and at 4 weeks after surgery.

15 CONCLUSION

Under the conditions of this test, the disposable cortical electrode was used to locate the brain functional areas. In the test process, the distribution of different limb parts in the brain functional areas could be reflected by the disposable cortical electrode. By performing recovery score and clinical observation, hematology and cerebrospinal fluid examination on all experimental animals, no abnormal lesion caused by the test articles were found. Histopathology results showed that the associated histopathological changes were minimal to mild neutrophil infiltration of the dura mater, leptomeninges, and arachnoid in the immediate postoperative period. Pathological changes were found in the brain tissue.

Combined with the accuracy of distinguishing brain functional areas, postoperative recovery, clinical observation and hematological examination, no long-term histopathological changes related to test articles occurred during the localization of brain functional areas, the feasibility, safety and effectiveness of judging the localization of brain functional areas were good.

16 ARCHIVE

The raw data (including electronic data), study documentation, specimens, the protocol, all amendments, and the original signed Final Report for this study shall be archived in the GLP archives according to Testing Facility SOPs. The electronic data and paper records collected in the test site were transferred to Testing Facility for archive. Following the contracted retention period, the Testing Facility will contact the Sponsor further disposition in accordance with Testing Facility SOPs. And Testing Facility reserves the right to retain exact copies (i.e., photocopies, scans, microfilm, etc.) for at least the minimum retention period specified by relevant regulations/principles. No materials will be discarded without the prior approval by the Sponsor.

17 REFERENCES

[1] Johnson P J, Luh W M, Rivard B C, et al. Stereotactic Cortical Atlas of the Domestic Canine Brain[J]. Scientific Reports, 2020, 10(1):4781.

This document contains proprietary confidential information of the Sponsor, Jiangxi NaoHu Technology Co., Ltd. and may not be disclosed verbally, abstracted or published without written consent of the Sponsor.

Confidential

Figure R4. Final report on the feasibility, safety, and effectiveness of the flexible electrodes in localizing functional brain areas of Labrador canine.

伦理审查批准函 (Ethics Review Approval Letter) form for the study on taste reconstruction in tongue cancer patients based on high-density flexible electrodes. Includes fields for study title, meeting location, sponsor, research institutions, principal investigator, review method, and documents of review.

伦理审查批准函 (Ethics Review Approval Letter) form for the study on taste reconstruction in tongue cancer patients based on high-density flexible electrodes. Includes review dates, review frequency, review frequency, and review frequency.

Figure R5. The ethics committee approval letter from Shanghai Ninth People's Hospital, Shanghai Jiao Tong University School of Medicine.

[4] The manuscript frequently mentions postoperative taste reconstruction, but the current research is limited to post-surgical sensing and decoding of TE signals (combined with EEG signals) without conducting clinical studies on active taste reconstruction based on postoperative taste assessments, such as feedback adjustment, stimulation, or pharmacological interventions. Therefore, consider whether the term "taste reconstruction" is appropriate.

We sincerely thank the reviewer for the valuable suggestion and we are sorry for our confusing use of the term. In this updated version of the manuscript, we have changed the title to "Gustatory Prosthesis for Operative Assessment and Taste Decoding in Patients with Tongue Cancer" to make it more consistent with the research work carried out in this article. We have revised all relevant usages of the term in the manuscript and supplementary information. We are aiming to emphasize that this work constitutes a foundational step towards the realization of real-time feedback and taste reconstruction in subsequent research endeavors.

[5] In experiments involving taste stimulation of subjects, detailed TE and EEG signals were recorded to construct a taste model. Should the subjective experiences of the subjects also be considered as part of the baseline or reference for the taste model, to better accommodate individual differences in taste decoding?

We thank the reviewer for the valuable comment. During the experiment, we initiatively elicited subjective feedback from the participants regarding their experience with each taste stimulus and utilized this information as a reference of the labels for taste decoding. However, due to the ambiguity in subjective experiences of taste stimulus intensity among many participants, obtaining reliable quantitative indicators proved challenging. Consequently, this subjective feedback was not utilized as a quantifiable input parameter for the model. In our study, we incorporated multiple participants to account for individual variations. Through normalization and other analytical methods, we confirmed the model's consistent and excellent performance across subjects, as shown in **Supplementary Figure 29** and **38** in revised manuscript.

[6] The study involves three preoperative patients and five postoperative patients. Were there any participants who took part in the clinical trials both before and after surgery? If so, please specify. If not, how does the constructed model account for individual differences among patients?

We thank the reviewer for the valuable comment. One of the patients met the predetermined inclusion criteria both before and after the surgical intervention, who was diagnosed with primary tongue cancer and underwent subsequent tongue flap resection and reconstruction surgery. We have validated the uniformity of the signals acquired from this patient before and after the surgery via the impedance of the electrodes and the intensity of the resting-state signals (**Figure R6a,b**). The TE signals evoked by taste stimulation from the natural tongue have also been inspected, showing similar time-domain waveforms and frequency-domain power distribution (**Figure R6c,d**). Minor differences may stem from the fact that the natural tongue is also affected by the transplantation surgery. Such uniformity substantiates the reliability of the recording.

To enhance the generalizability and robustness of our model across various time points and subjects, we systematically considered both patients at equivalent disease stages and diverse treatment stages of individual patients during the screening and enrollment processes.

Figure R6. Uniformity of acquisition from the same patient before and after the surgery. a, Impedance of the electrodes attached to the tongue surface of the patient preoperative and postoperative. **b,** The RMS values of the resting-state signal amplitude on the patient preoperative and postoperative. (ns denotes not significant, Mann Whitney test was used for **a,b**) **c,** Time traces of the representative TE signals evoked by taste stimulation acquired from the natural tongue preoperative and postoperative. **d,** Power spectral densities of the signals in **c**.

[7] How uniform are the TE signals measured by each electrode in the tongue electrode array? Has there been an effort to measure and establish a sensing baseline for each electrode? Furthermore, the manuscript uses normalized data to plot heat maps, such as Figure 3d. During normalization, is a unified baseline used, or is an independent baseline for each electrode applied, or perhaps a baseline derived from averaged processing?

We thank the reviewer for the valuable comment. On one hand, in the preparatory phase of each clinical trial, impedance and baseline signal measurements were conducted for each recording site of the tongue electrode array in a 1x phosphate-buffered saline (PBS) solution *in vitro*, as depicted in **Figure R7a**. Our tongue electrodes exhibited a narrow distribution of low impedance which determines the uniformity and effectiveness of the detected signals within the same environmental conditions, due to that low impedance is desirable to minimize noise and obtain high signal-to-noise ratio (SNR) recordings [Yu, K. J. et al. *Nat. Mater.* 15, 782–791 (2016); Thunemann, M. et al. *Nat. Commun.* 9, (2018); Qiang, Y. et al. *Sci. Adv.* 4, eaat0626 (2018); Jiang, Y. et al. *Science* 375, 1411-1417 (2022)]. The sensing baselines for each recording site and the power spectral densities have also been measured, which further confirmed the uniformity between channels (**Figure R7b,c**). On the other hand, in each clinical

trial we first performed impedance measurements for the electrode array attached to the tongue surface to ensure signal uniformity (**Figure R7d**). Additionally, sensing baselines were established for each recording site at the outset of every trial and the power spectra displayed a concentrated distribution (**Figure R7e,f**).

Given the potential variance in subjects' states and environments over an extended period, data normalization in **Figure 3d** in the original manuscript is achieved through max-min normalization utilizing a unified baseline for all electrodes within the same trial. Scaling the values of each electrode to a range between 0 and 1 facilitates intuitive comparison of the relative power intensity between the natural and reconstructed tongue flap at the two specified time points. Thanks again for the comment and we have clarified it in our revised manuscript.

Figure R7. Uniformity of the impedances and baseline signals for the tongue electrodes. RMS levels of baseline signals and impedances were estimated for all 256 electrode channels in 1x PBS (**a**) and on tongue surface (**d**). Sensing baselines for each recording site of the tongue electrodes in 1x PBS (**b**) and on tongue surface (**e**). Power spectral densities of sensing baselines in 1x PBS (**c**) and on tongue surface (**f**).

[8] In the synchronous measurement of the TE array, how is signal crosstalk between adjacent electrodes handled?

We thank the reviewer for this valuable comment. We agree that high-channel, high-density polymer electrodes pose a significant challenge on signal crosstalk resulting from the parasitic capacitance between the densely packed leads. In fact, before the design and fabrication of the electrode arrays, we have conducted finite element analysis (FEA) simulations to investigate the crosstalk between neighboring channels within an electrostatic field to provide guidelines for the design and optimization of the device. A 2D simulation model was built with two adjacent electrode traces embedded between the substrate layer and encapsulation layer (**Figure R8a**). An input voltage of 1V was applied to one trace and the other was grounded. We first simulated the crosstalk with different ratios between the gap and the width of the traces (W), which decreases with increasing ratio (**Figure R8b**). The dependence of the crosstalk on several

critical device parameters have also been explored (see **Figure R8c-f**). Increasing the gap with a given trace width helps mitigate the crosstalk, while the change of Au layer thickness (T_{Au}) has a slight impact on the crosstalk. As the thickness of the substrate (T_{sub}) or the insulation layer (T_{ins}) increase, the crosstalk decreases and eventually levels off. After comprehensive consideration of the crosstalk, the limitation of the fabrication process as well as the mechanical characterizations of the device, the relatively optimal parameters were selected to ensure that the crosstalk remains below 1% for neural recording [Qiang, Y. et al. *Nano Res.* 14, 3240-3247 (2021); Rios, G. et al. *Nano Lett.* 16(11), 6857-6862 (2016); Lopez, C. M. et al. *IEEE JSSCC* 49(1), 248-261 (2013)].

Figure R8. Simulation of the crosstalk. **a**, Simulation model with electrostatic potential mapping. **b**, Simulated crosstalk as a function of the gap between adjacent electrode traces in 1x PBS solution with different ratios of the gap to the width of the trace. Simulated crosstalk as a function of the trace gap (**c**), the thickness of Au trace (**d**), the thickness of the substrate (**e**), and the thickness of the insulator (**f**).

To further validate the anti-crosstalk performance of our electrodes, we have investigated the signal crosstalk between neighboring traces *in vitro*. The setup is shown in **Figure R9a** with

an aggressor trace connected to the waveform generator and a victim trace connected to the amplifier. The sinusoidal wave with a frequency of 20, 50, 100, 200, 500 and 1000 Hz was used as the test signal. Specific waveforms recorded from the aggressor and victim traces can be seen in **Figure R9b-g**, which suggest the same-frequency sinusoidal waves detected from the victim traces are much smaller than the aggressive signals. We summarized the crosstalk under different frequencies which were all well below 1% (**Figure R9h**).

Figure R9. *In vitro* evaluation of crosstalk between neighboring traces. **a**, The *in vitro* setup of crosstalk evaluation. Representative waveforms of the input sinusoidal signals (blue) to the aggressor trace with the frequency of 20 Hz (**b**), 50 Hz (**c**), 100 Hz (**d**), 200 Hz (**e**), 500 Hz (**f**), 1000 Hz (**g**) and the recorded signals (pink) acquired from the victim trace. Zoom-in views are enlarged waveforms detected from the victim trace. **h**, Summary of crosstalk under different frequencies.

It should also be noted that the crosstalk increases as the signal amplitude increases (take 50 Hz for example in **Figure R10**). The signal amplitude in the actual experiment is up to the hundred microvolt level, leading to the crosstalk that is far smaller than the observed test findings. This well satisfies the requirements of high-fidelity electrophysiological recording.

Figure R10. Crosstalk at various test signals with different peak-to-peak voltages under the frequency of 50 Hz.

Therefore, considering the above theoretical simulations and experimental results *in vitro*, we can draw a conclusion that the crosstalk is not a concern for our low impedance tongue electrodes. During subsequent data processing and analysis, we have removed all bad channels since any disconnected channel is equivalent to an antenna that only picks up noise and exhibits significant crosstalk [Tchoe, Y. et al. *Sci. Transl. Med.* 14(628), eabj1441 (2022)].

[9] As described by the authors, the collection of TE signals needs to be carried out multiple times both before and after surgery. Have the authors considered the tongue electrodes to be more durable "instruments" or easily replaceable "consumables"? If they are the former, how is the physical, chemical state, and performance stability of the tongue electrodes over time, and what is the data repeatability between multiple measurements? If they are the latter, how do you ensure the uniformity of signal measurements between electrodes prepared in the same or different batches?

We thank the reviewer for this insightful comment. Our tongue electrodes are designed to accommodate various application scenarios and the following elaboration will be developed from two perspectives.

On one hand, for multiple consecutive trials on the same patient, our tongue electrodes can be regarded as durable "instruments". Prior to conducting clinical trials, we investigated the stability of the tongue electrode by soaking it in 1x PBS for several weeks. Following five weeks of immersion, the structure and morphology of the electrode remained unaltered and exhibited initial flexibility and pliability, with no detachment and corrugation observed in the metal contacts (**Figure R11a**). In addition, the electrochemical impedance spectroscopy (EIS) in the frequency range of 1-100,000 Hz remained stable and the impedance at 1 kHz maintained at a constant value over the entire period (**Figure R11b,c**). We also characterized the noise level of the electrodes and observed that there were no significant changes in the noise of the signals (**Figure R11d**). These results validated the stability of our electrodes in aqueous media *in vitro*. Furthermore, to ascertain the performance stability of the electrodes in clinical trials, the impedance of the electrode attached on the tongue surface was measured over trials and remained constant as the experiment proceeded (**Figure R11e**). Given that the impedance of

the electrodes significantly influences the quality of signals, the noise maintained at a relatively stable and low level across multiple trials, ensuring good repeatability between multiple measurements (**Figure R11f**). These findings *in vivo* suggest that the durability of our tongue electrodes contributes to form a conformal, stable and reliable bioelectronic interface.

Figure R11. Performance stability of the tongue electrodes. **a**, Photograph of the tongue electrode after 5 weeks of immersion in 1x PBS. Inset: zoomed-in microscope image of the metal contact without detachment and corrugation. **b**, Electrochemical impedance spectroscopies of the tongue electrode soaked in 1x PBS solution for 5 weeks. **c**, Impedance stability of the tongue electrode at 1 kHz over time when immersed in 1x PBS solution. **d**, Characterization of the noise level showing good stability in the ability to acquire signals of the electrode incubated in 1x PBS solution for 5 weeks. **e**, Impedance changes of the electrode on tongue surface over trials. **f**, The relatively stable and low noise level across multiple trials.

On the other hand, for different subjects before and after surgery, brand new tongue electrodes were prepared in accordance with the security requirements of clinical trials and to prevent cross-infection between patients. From this point of view, our electrodes are indeed

replaceable “consumables”. Drawing on the batch fabrication techniques exploited by the integrated circuit (IC) industry, the electrodes are fabricated using the standard micro-electro-mechanical systems (MEMS) process for semiconductor integrated manufacturing, which contributes to a low total defect count [Judy, J. W. *Smart Mater. Struct.* **10**, 1115 (2001); Fischer, A. C. et al. *Microsyst. Nanoeng.* **1**, 1-16 (2015); Hajare, R. et al. *Mater. Today: Proceedings* **49**, 720-730 (2022)]. To verify the uniformity and reproducibility, three samples (Labeled as Sample 1, Sample 2, and Sample 3) are randomly selected for control experiments. Specifically, the sample 1 and 2 are prepared in the same batch while the sample 3 is chosen from another batch. The EIS and power spectral density (PSD) of all recording sites from three samples in 1x PBS buffer are measured and compared in **Figure R12a,b**, which are almost overlapped, indicating high consistency in terms of the electrochemical characterization and electrical performance. **Figure R12c** shows the distribution of the impedance and phase at 1 kHz and the RMS value of the signal in 1x PBS buffer which correspond to x-, y-, and z-direction separately. A smaller area signifies denser data distribution, reflecting a higher level of uniformity among the electrodes prepared in the same or different batches. The relative standard deviations (RSD) across the same or different batches are calculated to quantify the uniformity of signal measurements (**Figure R12d**), which suggest the narrow fluctuation of the performance and the relatively stable manufacturing process.

Figure R12. Uniformity and reproducibility of electrical performance and signal measurements between electrodes prepared in the same and different batches. a, EIS plots for all recording sites from the electrode sample 1, 2, and 3 in 1x PBS buffer. **b,** PSD for all recording sites from the electrode sample 1, 2, and 3 in 1x PBS buffer. **c,** The comparisons of impedance (at 1 kHz), phase (at 1 kHz), and RMS noise of the signals for different electrodes. **d,** RSD between the electrodes in the same batch and from different batches.

[10] It is mentioned that 0.5 M solutions of five different tastes were used as taste stimuli. However, based on the typical ranges for daily diets, as well as the literature cited in references 30 and 31, the concentration of 0.5 M for citric acid and magnesium chloride to provide sour and bitter stimuli appears to be excessively high. It may even cause discomfort for the subjects. Similarly, the concentration of 0.5 M glutamate sodium for umami taste is also relatively high. Could the authors explain the rationale behind using such high concentrations of taste stimuli solutions and its impact on the experimental results?

We thank the reviewer for raising this valuable concern, and apologize for any lack of clarity in the original manuscript. Compared to quinine monohydrate used in references 30 and 31 with an extremely bitter taste, which also acts as an antimalarial, anticholinergic, antihypertensive and a hypoglycemic agent, we chose a safer and more acceptable alternative – magnesium chloride (the main component of the food additive brine) adopted in references [Wang, H. et al. *Sensors* 21, 6965 (2021); Miyashita, H. 2020 *CHI Conference on Human Factors in Computing Systems* (2020)]. The concentrations of 0.5 M for sodium chloride, sucrose, sodium glutamate, and magnesium chloride are investigated in the article [Wang, H. et al. *Sensors* 21, 6965 (2021)]. A variety of factors such as age, genetic polymorphisms, and living habits give rise to large inter-individual differences in taste thresholds [Barragán, R. *Nutrients* 10(10), 1539 (2018); Wiriyawattana, P. et al. *J.Sens.Stud.* 33(4), e12436 (2018)]. We have taken into account the fact that our subjects were screened from dental outpatients, covering a wide range of ages and genders. Besides, the scope of this research focuses on the “yes” or “no” of taste perception. Although we recognize that the concentration for citric acid appears to be high compared to previous studies, the concentration of 0.5 M for all five tastes was chosen to maintain uniformity of the objective variables in the experiment and to be more clearly perceivable for all participants. More importantly, no subjects reacted with disgust or discomfort throughout the whole clinical trials.

To validate the impacts of different concentrations on the experimental results, we compared the decoding performance of TE and EEG signals evoked by four groups of taste stimuli solutions with different concentrations, especially including 0.039 M adopted in the reference 30 and 31 for citric acid (**Figure R13**). Results show that such high concentrations of taste stimuli solutions do not make the decoding less accurate.

Figure R13. Effects of taste stimuli solutions with different concentrations on decoding performance. Comparison of the decoding accuracy from four groups of taste stimuli solutions with different concentrations. Noted that the concentration of 0.039 M was adopted in the reference 30 and 31 for citric acid and 0.5 M was used in our manuscript.

[11] The authors have opted for time-domain features due to their computational ease. While this choice is justified for initial studies, the complexity of biological signals might be better represented using a combination of both time and frequency-domain features. It would be beneficial to discuss the potential advantages or results of incorporating such features and whether any preliminary investigations in this direction were conducted.

We thank the reviewer for the valuable comment. We agree that combining both time and frequency-domain features can offer a more comprehensive view of the biological signals and have the potential to enhance the performance of machine learning algorithms in classifying and discriminating between different physiological states. Before finalizing the decoding framework, we have conducted a comparative analysis to ascertain the optimal decoding performance, which involved evaluating decoding accuracy through the extraction of various time-domain and frequency-domain features, as well as their combinations. The specific frequency-domain features are listed in **Table 1**. The features of different domains and their specific combinations were validated on individual patients (**Table 2**) and across five subjects (**Figure R14**). The results show that time-domain features contribute to the highest accuracy across subjects despite insignificant differences on individual patients. We posit that the observed outcomes can be attributed to the substantial inter-subject variability exhibited by frequency-domain features, whereas time-domain features manifest more pronounced commonalities within our specific application scenario. Here we chose the time-domain feature

Ku (Kurtosis, outlined with dotted red lines) with smaller inter-subject variance, while the computational efficiency makes them suitable for future real-time processing and wireless, low-power devices.

Table 1. Three investigated frequency-domain features

No.	Frequency-domain Feature	Equation
1	Mean Frequency (MNF)	$MNF = \frac{\sum_{j=1}^M f_j P_j}{\sum_{j=1}^M P_j}$
2	Root Mean Square Frequency (RMSF)	$RMSF = \sqrt{\frac{\sum_{j=1}^M (f_j)^2 P_j}{\sum_{j=1}^M P_j}}$
3	Standard Variance Frequency (RVF)	$RVF = \sqrt{\frac{\sum_{j=1}^M (f_j - f_c)^2 P_j}{\sum_{j=1}^M P_j}}$

Table 2. Accuracy on individual patients

	Ku	MNF	Ku+MNF	RMSF	Ku+RMSF	RVF	Ku+RVF
PA1	100	96.00	96.00	96.00	96.00	96.00	96.00
PA2	100	100	100	100	100	100	100
PA3	100	100	100	100	100	100	100
PA4	100	86.67	86.67	100	86.67	86.67	100
PA5	100	100	100	93.33	100	100	100

Note that Ku (Kurtosis) is the time-domain feature adopted in the original manuscript.

Figure R14. Comparative analysis of cross-patient decoding accuracy on the gustatory information of the reconstructed tongue using time-domain, frequency-domain features and their combinations.

[12] Linear Discriminant Analysis (LDA) is chosen for classifying taste signals. Although LDA is known for its simplicity and efficiency, it might not be the most robust option for complex signal classification tasks. Consideration of non-linear classifiers, such as Support Vector

Machines, Random Forests, or even deep learning models, could potentially enhance classification performance. A comparative study of these models against LDA, with relevant performance metrics, would provide a more comprehensive understanding of the classifier's suitability.

We thank the reviewer for the valuable comment. We agree that non-linear classifiers like Support Vector Machines (SVM) and Random Forests (RF) can model complex relationships between features and the target variables and have the potential to improve prediction accuracy. Therefore, we have compared the performance of multiple classifiers, including LDA, SVM, RF, DT (Decision Tree) and NB (Naïve Bayes), in order to select a more appropriate and suitable one. The prediction accuracy of the gustatory information decoded from the reconstructed tongue and the corresponding time consumption on each subject are shown in **Figure R15a,b**. The LDA, SVM, and RF demonstrate favorable performance concerning classification accuracy for five patients, with LDA being the most time efficient. Furthermore, our comparative analysis on cross-subject datasets reveals that LDA exhibits not only the highest decoding accuracy but also demonstrates the shortest decoding time relative to other methods examined (**Figure R15c,d**). In the current application scenario, considering a comprehensive assessment encompassing decoding accuracy and computational efficiency, LDA emerged as the preferred choice. While non-linear classifiers may offer advantages in terms of their ability to capture complex relationships, flexibility in decision boundaries, and robustness to noise and outliers, they usually require more computational time and resources for training and inference and tuning of hyperparameters to achieve optimal performance, which limit their applicability in resource-constrained environments or real-time systems.

Figure R15. Comparative analysis of performance metrics between different classifiers, including LDA, SVM, RF, DT, and NB. a, Decoding accuracy of the gustatory information from the reconstructed tongue utilizing different classifiers on individual patients (PA1, PA2, PA3, PA4,

and PA5). **b**, Decoding time using different classifiers on individual patients. **c**, Cross-patient decoding accuracy of the gustatory information from the reconstructed tongue utilizing different classifiers. **d**, Decoding time using different classifiers on cross-patient datasets.

Reviewer #2:

[1] The authors purport to have developed a novel device for detecting and directing surgical management of lingual cancer. This device is a thin, flexible, 256 electrode array that is applied to the surface of the tongue and that is designed to record taste-evoked responses from the lingual surface. In addition to being designed to assess lingual cancer, the authors claim that the device has the potential to enhance taste by combining their technology with EEG recordings (brain computer interface, BCI). Unfortunately, the report has several flaws that lead one to question the significance and relevance of the findings.

We sincerely thank the reviewer for the time involved in reviewing the manuscript and appreciate the valuable comments.

[2] First, the authors claim to have recorded taste-evoked responses from the electrode array (Fig 3c-e). The figure appears to show data from one patient, 3 and 9 months after surgical removal of lingual cancer, although the details are scanty. Thus, the author's statement about Statistical Analysis was unconvincing ("All experiments were conducted with a minimum of $N = 3$ for each data point").

We thank the reviewer for raising this important issue and apologize for any lack of clarity in our original manuscript. This statement refers to the statistical test involved in our manuscript, such as **Figure 3e** (**Figure 3d** in revised manuscript) and **Supplementary Figure 16**. The significant differences were obtained with $N \geq 3$. More importantly, to make our findings more convincing, the other two postoperative patients underwent another follow-up examination and the results have been added to the **Figure 3** of the revised manuscript. Through longitudinal monitoring of three patients after surgery, it can be observed that as the recovery time increases, the TE signals of the reconstructed tongue gradually approach those of the natural part in terms of the spatial distribution and quantitative power ratio. Thus, the results from three patients validate that the tongue electrical signals can serve as indicators of tongue recovery and our gustatory prosthesis enables comprehensive monitoring through the postoperative period.

Figure 3 | Postoperative tongue structure recovery monitoring. **a**, Schematic illustration depicting the application scenario for structure recovery monitoring. The TE signals are acquired from the reconstructed and natural tongue of a patient who has undergone resection and reconstructive surgery. **b**, Spectrum of TE signals from the natural and reconstructed tongue. The inset displays representative 10-second time traces from these two areas, with the first five seconds in a relaxed state and the second half in a tense state. **c**, Normalized power heatmaps of taste-induced TE signals of three patients (PA1, PA2 and PA3) during two follow-up visits after surgery. The examinations were conducted at one and twelve months after surgery for PA1, at two and fifteen months after surgery for PA2, at three and nine months after surgery for PA3. The boundary between the reconstructed and natural parts of the tongue is demarcated with dotted lines. For PA1 and PA3, the left side of the dotted line refers to the reconstructed tongue while the right side of the line is the reconstructed tongue for PA2. **d**, Normalized power ratio between the reconstructed and the natural tongues under five taste stimulations for three patients during two follow-up examinations after surgery. ($*p < 0.05$, $**p < 0.01$, $***p < 0.001$, $****p < 0.0001$, two-way ANOVA).

[3] Moreover, the taste stimuli are highly unusual and inappropriate. Namely, for testing taste-evoked responses, Wang et al applied solutions of 500 mM NaCl (salty), 500 mM sucrose (sweet), 500 mM citric acid (sour), 500 mM sodium glutamate (umami) or 500 mM magnesium chloride (bitter). The authors claim these stimuli and these concentrations those used in other studies (refs 30, 31). However, only NaCl and sucrose stimuli were similar to those published in refs 30, 31. Contrary to the authors' claim, Wang et al applied much higher concentrations of taste stimuli than used in the publications they cited (e.g. the cited publications elicited sour with 39 mM citric acid and bitter with 0.2 mM quinine), or not tested at all (sodium glutamate). We thank the reviewer for raising this valuable concern, and apologize for any lack of clarity in the original manuscript. Compared to quinine monohydrate used in references 30 and 31 with an extremely bitter taste, which also acts as an antimalarial, anticholinergic, antihypertensive and a hypoglycemic agent, we chose a safer and more acceptable alternative – magnesium chloride (the main component of the food additive brine) adopted in references [Wang, H. et al. *Sensors* 21, 6965 (2021); Miyashita, H. 2020 *CHI Conference on Human Factors in Computing Systems* (2020)]. The concentrations of 0.5 M for sodium chloride, sucrose, sodium glutamate, and magnesium chloride are investigated in the article [Wang, H. et al. *Sensors* 21, 6965 (2021)]. A variety of factors such as age, genetic polymorphisms, and living habits give rise to large inter-individual differences in taste thresholds [Barragán, R. *Nutrients* 10(10), 1539 (2018); Wiriyawattana, P. et al. *J. Sens. Stud.* 33(4), e12436 (2018)]. We have taken into account the fact that our subjects were screened from dental outpatients, covering a wide range of ages and genders. Besides, the scope of this research focuses on the “yes” or “no” of taste perception. Although we recognize that the concentration for citric acid appears to be high compared to previous studies, the concentration of 0.5 M for all five tastes was chosen to maintain uniformity of the objective variables in the experiment and to be more clearly perceivable for all participants. More importantly, no subjects reacted with disgust or discomfort throughout the whole clinical trials.

To validate the impacts of different concentrations on the experimental results, we compared the decoding performance of TE and EEG signals evoked by four groups of taste stimuli solutions with different concentrations, especially including 0.039 M adopted in the reference 30 and 31 for citric acid (**Figure R16**). Results show that such high concentrations of taste stimuli solutions do not make the decoding less accurate. We appreciate the reviewer's concern and we will remain committed to further investigation and exploration.

Figure R16. Effects of taste stimuli solutions with different concentrations on decoding performance. Comparison of the decoding accuracy from four groups of taste stimuli solutions with different concentrations. Noted that the concentration of 0.039 M was adopted in the reference 30 and 31 for citric acid and 0.5 M was used in our manuscript.

[4] Second, and more importantly, the electrical activity of taste buds and/or taste afferents (Fig 2C) were unconvincing. There is no marking for when the taste stimuli were applied and no way to differentiate electrical signals from other sources (thermal, tactile, or more likely, EMG). Moreover, the topography of the electrical signals (Fig.2D) doesn't appear to reflect any particular distribution of taste buds.

We thank the reviewer for the valuable comments and we are sorry for any confusion related to **Figure 2c** and **2d**. Before starting signal acquisition in each clinical trial, the impedances were measured and evaluated for each recording site of the electrode array since the recording quality is systematically related to electrode impedance [Lewis, C. et al. *Adv. Healthc. Mater.* 2303401 (2024); Chung, T. et al. *J. Neural. Eng.* 12(5), 056018 (2015)]. **Figure 2c** shows the impedance distribution of the 256 channels in our tongue electrode in the form of a histogram, from which it can be observed that the impedance of all channels is well below 100 k Ω . The inset heatmap is depicted by mapping the impedance of each channel to its actual location to visualize the spatial distribution rather than the electrical activity of taste buds and/or taste afferents. The uniformity of low impedance ensures the effectiveness of signals and the credibility of subsequent analyses.

Regarding the marking of the taste stimuli, we have recorded the onset and end of the applied stimulus corresponding to our experimental paradigm which we mentioned in the

Methods in our manuscript.

With respect to the interference with electrical signals from other sources, we have taken a series of measures to eliminate interference from extraneous signals during the experiment and subsequent data processing. The interface impedance instability motion artefacts can be well managed by virtue of the stable conformability at the electrode-tongue interface while other mechanical motion artefacts can be addressed by algorithmic intervention. Take thermal interference as an example, almost no temperature variations were introduced during our experiments and the electrodes are temperature-insensitive in the TE acquisition environment, so that interference from thermoelectric signals can be excluded. Then tactile-induced electrical signals are a kind of mechanical motion artefacts with different characteristics compared to our targeted electrical signals. Therefore, the signals can be post-processed employing advanced filtering algorithms and machine learning models to separate the target signals from extraneous artefacts, thus enhancing the fidelity and precision of the extracted target signals. [Yin, J. et al. *Nat. Rev. Bioeng.* 1-18 (2024)].

Tongue is a special organ that contains a large number of cells and neurons. The taste receptor cells on the tongue can be excited by the chemical components, resulting in the discharge of the taste cells and the generation of electrical activities. Meanwhile, tongue acts as a kind of muscular organ which composes of various muscle fibres [Stål, P. et al. *Cells Tissues Organs* 173(3), 147-161 (2003)]. Tongue muscle undergoes potential changes during excitation due to conduction and spreading of action potentials in muscle fibers. Furthermore, taste information is transmitted by cranial nerves to gustatory and sensory cortices through nuclei in the brainstem and thalamus, while one of cranial nerves, chorda tympani branch of the facial (CN VII), enters the anterior 2/3 of the tongue [Doyle, M. E. et al. *Physiol. Rev.* 103(2), 1193-1246 (2023)]. Thus, the electrical signals acquired through our tongue electrodes are the result of the superposition of multiple cellular and neural electrical activities, which forms the basis for the gustatory decoding of the reconstructed tongue.

We apologize again for any confusion related to **Figure 2d**. The top figure shows a preoperative CT image of the tongue from one patient and the gray dots mark the locations of the 256 electrode channels. The bottom highlights the time-domain waveforms of the electrical signals from the cancer central site and a small area around it (black dashed box). These results aim to illustrate the difference in electrical signals between the tongue cancer site and the surrounding normal tissues, rather than the distributions of taste buds. With respect to the distribution of taste buds, we have investigated the spatial mapping of the tongue electrical signals evoked by five taste stimulations in **Supplementary Figure 11e**. The results show that there is no clear and distinguishable distribution of electrical activities, which is an indirect reflection of the taste bud distribution. Such conclusion is in line with the current opinion that responsiveness to the five basic modalities — bitter, sour, sweet, salty and umami — is present in all areas of the tongue [Chandrashekar, J. et al. *Nature* 444, 288–294 (2006); Spence, C. et al. *Curr. Opin. Food. Sci.* 5, 598-610 (2022)].

[5] Wang et al did not explain the significance of “power” and “frequency spectrum” in the recordings very well (Fig 2b,e,f,g,h; Fig 3b,c,d,e). If these are significant measurements, the authors should describe and discuss what the data signify.

We thank the reviewer for the valuable comment and sorry for any confusion and unclarity. The

first we want to note that **Figure 2b** depicts the normalized total energy landscape of the flexible tongue electrodes laminating on the tongue surface, which can be used to predict conformability conditions. The normalized total energy is plotted versus \hat{x}_c (the degree of conformability) and ξ (the degree of deformation of the tissue) according to the following function with all other parameters fixed [Wang, L. et al. *J. Appl. Mech.* **83(4)**, 041007 (2016)]:

$$\hat{U} = \alpha \frac{\xi^2}{12} \eta^3 D(\hat{x}_c) + \alpha \eta \xi^4 \beta^2 K(\hat{x}_c, \xi \beta) - \frac{\mu}{\beta^2} E(\hat{x}_c, \xi \beta) + \frac{(1 - \xi)^2}{4\pi} [F_1(\hat{x}_c) - F_2(\hat{x}_c)]$$

Minimization of the normalized total energy can give us the equilibrium solution which is visualized as the global minimum of the 3D plot (highlighted as the red dot). The theoretical calculations show that the thickness of 20 μm corresponds to the full conformability (\hat{x}_c is pretty close to 1), which is in agreement with the experimental results.

The power and frequency spectrum of the tongue electrical signals are obtained by spectral analysis methods, which are now routinely used in electrophysiological studies [Gao, R. J. *J. Neurophysiol.* **115(2)**, 628-630 (2016); Donoghue, T. et al. *Nat. Neurosci.* **23**, 1655–1665 (2020); Voytek, B. et al. *Biol. Psychiat.* **77(12)**, 1089-1097 (2015)]. Inspired by the approaches adopted in the analysis of local field potential (LFP), electrocorticogram (ECoG), and electroencephalogram (EEG), we have characterized the tongue electrical signals with frequency-domain representations to investigate the relationship between tongue electrophysiology and diseases. The time-frequency spectra in **Figure 2e** and **2f** reveal fluctuation patterns of tongue electrical activities, like neural oscillations, to identify disease-related modulations. The power spectral density (PSD) in **Figure 2g** describes how the energy of the tongue electrical signals from the normal tissues and cancer site is distributed with frequency. From the above analysis, it can be observed that in the specific frequency band of 10-500 Hz, there exists a difference in terms of the signal power between the normal tissues and cancer site. Thus, the root mean square of amplitude has been calculated as the power and mapped into the actual position to visualize the cancer site localization (**Figure 2h**). Similar to **Figure 2g**, the PSD in **Figure 3b** shows the strength of the tongue electrical signals from the natural and reconstructed tongue without taste stimuli as a function of frequency. Fig. 3c (**Supplementary Figure 15** in revised manuscript) depicts the time-domain waveforms and corresponding time-frequency spectra, while the spatial distributions and ratios of the normalized root mean square of amplitude are shown in **Figure 3d** and **3e** (**Figure 3c** and **3d** in revised manuscript). The power and spectrum are both a measurement of the intensity of cellular discharge activities, any abnormalities of which is likely to be related with alterations in physiological structures. We have added some discussion in our revised manuscript and hope that this clarification improves the understanding of our analysis.

[6] Fourth, the potential use for such a device in a clinical environment is dubious. Oral cancer patients experience significant mechanical allodynia. Applying and using a prosthetic device such as Wang described, regardless of how soft and pliable is the array, is likely to be quite painful and rejected. The authors did not discuss this aspect of their device.

We thank the reviewer for raising this important concern. We admit that our tongue electrodes may cause a certain amount of foreign body sensation, but as a non-invasive means, it does not cause any pain to the patient. Prior to the clinical trial, we will ensure each participant's informed consent to the experimental procedure and the devices used. There was no patient

feedback of discomfort during or after the experiment. All these guaranteed the smooth running of our experiment. Although such a device is still immature for clinical use, we would like to emphasize that it offers a novel proof-of-concept idea for surgical evaluation, postoperative monitoring and decoding of taste function in patients with tongue cancer. Requirements for applications in clinical scenarios suggest future research directions, including wireless bioelectronic interface. We have added the discussion in the section Discussion and Conclusion in our revised manuscript. Thank again for the reviewer to help us clarify this important aspect of our work.

REVIEWER COMMENTS

Reviewer #1 (Remarks to the Author):

The authors have adequately addressed all my concerns; however, I still have some doubts regarding the response to question 3. The authors have presented comprehensive test reports and ethical approval for the clinical study, but the "product name" in these documents (including the documents for the ethical review) are all labeled as "disposable electrocorticography (ECoG) electrode", both in Chinese and English. Therefore, I am curious whether the authors used a commercially available ECoG electrode as the tongue electrodes in this study, or at least as a prototype. Yet, the authors claim that the tongue electrodes was specifically designed and manufactured in the manuscript. Please clarify it.

Reviewer #3 (Remarks to the Author):

First, we do not believe this gustatory "prosthesis" represents a true prosthesis, since it does restore function of the missing receptors in the way that other sensory prostheses, such as cochlear implants, do. We thought originally after reading the title that the electrodes making up the prosthesis would directly detect patterns of activity from the taste stimuli and transmit these signals to the gustatory afferents to restore taste function, in a manner analogous to cochlear implants restoring hearing to patients that have damaged or lost sensory hair cells. However, this prosthesis simply provides a gustatory readout of taste function from the existing epithelium that still contains taste buds. Although the prosthesis provides a detailed readout of taste recovery in different regions of the tongue, it is not clear that it provides any help to the patients in identification of the different tastes during recovery, even when coupled with the EEG recordings.

Second, it is not clear that loss of taste acuity is a significant problem with tongue cancer patients. In doing a cursory search of the literature, we encountered a manuscript by Shibahara et al., "Evaluation of taste sensation following tongue reconstruction by microvascular forearm free flap", *J. Oral Maxillofac Surg* 63:618-622, 2005. In this paper, which was not cited by the authors, taste recovery was examined in 12 patients with free flap reconstructions of the tongue regions, using filter discs impregnated with reasonable concentrations of sweet, salty, sour and bitter stimuli. Although no taste function was observed directly under the reconstructed area, similar to the results in the present paper, when taste stimuli were applied to the whole tongue the authors concluded "none of our postsurgery patients showed any serious taste disorder". They suggested that compensatory mechanisms in the unoperated side of the tongue contribute to restoring taste function. These findings diminish the significance of the present paper and suggest the authors did not do a very scholarly assessment of the existing literature. In going through the points raised by the previous two reviewers, we can really only comment on the points raised that relate to taste perception-- the technical details concerning the operation and testing of the device are beyond our expertise. However, just looking at their responses to questions relating to taste stimuli, we completely agree with the concerns about taste stimuli concentrations. The authors did not adequately justify their use of high concentrations of citric acid and why they chose MgCl₂ as the bitter taste stimulus. MgCl₂ is both bitter and salty and is thus not a pure stimulus. Quinine is the classic bitter stimulus and is completely non-toxic and very bitter at concentrations as low as 10 mM. Since the patients are not ingesting the stimuli, the fact that quinine is used clinically should not be relevant. Also

the authors did not adequately explain how they differentiated taste signals from the many mechanosensory signals that arise from the tongue.

Although the prosthesis does provide a detailed map of the tongue in response to the different taste qualities and could be useful for certain applications, it does not reach the level of significance that we would expect of a Nature Communications manuscript. I think it would be more appropriate in a specialty journal, perhaps an engineering journal

Point by point response (comments in black and responses in blue):

Reviewer #1 (Remarks to the Author):

[1] The authors have adequately addressed all my concerns.

We sincerely thank the reviewer for the positive comment.

[2] However, I still have some doubts regarding the response to question 3. The authors have presented comprehensive test reports and ethical approval for the clinical study, but the "product name" in these documents (including the documents for the ethical review) are all labeled as "disposable electrocorticography (ECoG) electrode", both in Chinese and English. Therefore, I am curious whether the authors used a commercially available ECoG electrode as the tongue electrodes in this study, or at least as a prototype. Yet, the authors claim that the tongue electrodes was specifically designed and manufactured in the manuscript. Please clarify it.

We thank the reviewer for the valuable comments. First, we would like to clarify that the tongue electrodes are designed in-house rather than adopting a commercially available ECoG electrode. We performed fabrication on our own R&D line subsequently to verify the feasibility of the process we designed and the functional effectiveness of the electrodes. Second, in order to make the tongue electrodes biologically safe to meet the requirements for clinical use, we have commissioned a third-party foundry which possesses a national quality management system-certified MEMS pilot R&D platform and ultra-clean rooms of a higher purification level to finish the fabrication. Last, the biocompatibility and safety of a device are primarily determined by its material and manufacturing process, while the tongue electrode in this manuscript shares identical materials and manufacturing processes with the "disposable ECoG electrode" we previously developed, differing only in device shape. Thus, we used test reports of the disposable ECoG electrodes to corroborate the biocompatibility of the tongue electrodes.

Reviewer #3 (Remarks to the Author):

[1] First, we do not believe this gustatory “prosthesis” represents a true prosthesis, since it does not restore function of the missing receptors in the way that other sensory prostheses, such as cochlear implants, do. We thought originally after reading the title that the electrodes making up the prosthesis would directly detect patterns of activity from the taste stimuli and transmit these signals to the gustatory afferents to restore taste function, in a manner analogous to cochlear implants restoring hearing to patients that have damaged or lost sensory hair cells.

We thank the reviewer for the valuable comment and apologize for any confusion. We fully agree with the reviewer that the original concept of neuroprosthetics involves the artificial manipulation of the biological neural system using externally induced electrical currents, aimed at restoring damaged sensorimotor functions, and that the cochlear implant exemplifies the most widely used neuroprosthesis. With advancements in neuroengineering, brain-machine interfaces (BMIs), an artificial process that facilitates direct communication between the brain and external devices, have emerged as a new generation of neuroprosthetic devices with the same goal of assisting, augmenting, or repairing sensorimotor or cognitive functions¹⁻².

Recent research on speech and motor neuroprostheses has demonstrated their capability to decode cortical activity associated with attempted speech and voluntary motor impulses into text or command inputs for digital devices³⁻⁶. Building upon the concept of such neuroprostheses, which bypass damaged neural circuits and establish alternative pathways to enable communicative or motor functions, we apply the term “prosthesis” to our work which involves decoding gustatory information from electrophysiological signals recorded from reconstructed tongue flaps and EEG signals.

In the revised manuscript, we have replaced the term “prosthesis” with “interface” to make it more accurate in summarizing the entire work. And we have revised all relevant usages of the term in the manuscript and supplementary information.

[2] However, this prosthesis simply provides a gustatory readout of taste function from the existing epithelium that still contains taste buds. Although the prosthesis provides a detailed readout of taste recovery in different regions of the tongue, it is not clear that it provides any help to the patients in identification of the different tastes during recovery, even when coupled with the EEG recordings.

We thank the reviewer for the valuable comments and apologize for not adequately summarizing the key contributions and highlights of our work. Following an extensive review of the literature on preoperative and postoperative taste perception in patients with tongue cancer, we found that electrophysiological studies specifically related to reconstructed tongue flaps remain less explored. Our study not only provides a detailed taste mapping between natural and reconstructed tongue regarding postoperative recovery, but more significantly, our research introduces an innovative approach by accurately predicting gustatory information from electrophysiological signals of the reconstructed free flap, which lacks taste buds, achieved through advanced multimodal recording and decoding algorithms. This approach could help patients differentiate flavors from unfamiliar and unknown foods.

[3] Second, it is not clear that loss of taste acuity is a significant problem with tongue cancer patients. In doing a cursory search of the literature, we encountered a manuscript by Shibahara

et al., “Evaluation of taste sensation following tongue reconstruction by microvascular forearm free flap”, J. Oral Maxillofac Surg 63:618-622, 2005. In this paper, which was not cited by the authors, taste recovery was examined in 12 patients with free flap reconstructions of the tongue regions, using filter discs impregnated with reasonable concentrations of sweet, salty, sour and bitter stimuli. Although no taste function was observed directly under the reconstructed area, similar to the results in the present paper, when taste stimuli were applied to the whole tongue the authors concluded “none of our postsurgery patients showed any serious taste disorder”. They suggested that compensatory mechanisms in the unoperated side of the tongue contribute to restoring taste function. These findings diminish the significance of the present paper and suggest the authors did not do a very scholarly assessment of the existing literature.

We thank the reviewer for the insightful comment. During our literature review and evaluation, we have read the manuscript mentioned by the reviewer and the conclusion regarding the deficiency of taste perception in the tongue regions subjected to reconstructive surgery with forearm flaps constitutes a significant foundation for our study. The investigation of taste perception restoration in patients who have undergone tongue reconstruction with microvascular flaps harvested from the forearm following ablative surgery can serve as a critical prognostic indicator of postoperative outcomes and facilitate the advancement of more effective reconstruction techniques. This finding is consistent with our clinical observations, where the patients who underwent approximately half of the tongue flap excision experienced mild taste abnormalities. However, this was not the case for patients with a larger area of tongue removal, who exhibited more significant taste impairment. We apologize for the lack of comprehensiveness in citing references and we have included this paper in our revised manuscript.

On the other hand, we have included an investigation of recent literature that supports the aforementioned conclusions⁷⁻¹². Also, these studies highlight that the severity of dysgeusia is influenced by multiple factors, including the extent of surgical resection, the integrity and disruption of the lingual nerve, and the methods employed for reconstruction. Notably, a greater extent of tongue resection is associated with increased damage to the glossopharyngeal nerve, resulting in diminished taste acuity and its detection threshold.

Some of references and citations supporting the above speculation are listed as followings:

*“The extent of surgical resection and the reconstruction modalities can affect the sensation of taste. Preserving more than the 50% of the tongue base, taste acuity and its detection threshold are improved. Additionally, reconstruction modalities influence taste perception. Free flaps, regardless of the type, significantly improve dysgeusia compared to no-free flaps.”*⁷

*“Eleven of the 24 patients were aware of their taste disorder after surgery. Four patients with more than 1/2 residual tongue base had no taste complaints, whereas seven of 14 patients with less than 1/3 residual tongue base reported taste abnormalities.”*⁸

*“The primary results of this study revealed that the patients with lingual nerve reconstruction met with modest success in restoring sensation, and that leaving the lingual nerve severed resulted in oral sensory impairments. When compared to matched controls, all patients with lingual nerve disruption exhibited significantly poorer outcomes for taste. Patients with the lingual nerve left intact had superior outcomes in sensory function comparable to function in the matched controls.”*⁹

In line with this, our clinical cohort demonstrated that patients who had two-thirds or more of the tongue flap removed experienced significant difficulty in distinguishing between different flavors and frequently expressed dissatisfaction and complaints regarding their taste disorder. Therefore, our proposed method aims to offer a novel approach for decoding gustatory information from tissues lacking taste buds, providing a potential solution for patients experiencing taste loss due to extensive tongue flap resection and reconstruction.

[4] In going through the points raised by the previous two reviewers, we can really only comment on the points raised that relate to taste perception-- the technical details concerning the operation and testing of the device are beyond our expertise. However, just looking at their responses to questions relating to taste stimuli, we completely agree with the concerns about taste stimuli concentrations. The authors did not adequately justify their use of high concentrations of citric acid and why they chose $MgCl_2$ as the bitter taste stimulus. $MgCl_2$ is both bitter and salty and is thus not a pure stimulus. Quinine is the classic bitter stimulus and is completely non-toxic and very bitter at concentrations as low as 10 mM. Since the patients are not ingesting the stimuli, the fact that quinine is used clinically should not be relevant.

We thank the reviewer for the valuable comments. On the one hand, the use of magnesium chloride ($MgCl_2$) as a bitter stimulus is supported by several studies related with taste sensation and receptors (other than reference 33 and 34)¹³⁻¹⁵. On the other hand, safety and patient acceptance are critical considerations that need to be prioritized in the design of experimental paradigms. For this reason, we opted for the food-grade additive – $MgCl_2$ to simulate taste experiences that closely resemble those encountered in daily life. Quinine is primarily used as a drug for the treatment of malaria, lupus, and other diseases. We provided patients with the option to choose between quinine and $MgCl_2$ as bitter stimuli and informed them of the details of both solutions during enrollment for clinical trials, and all patients selected $MgCl_2$ as the testing reagent. We have included the explanations in the Methods section of the revised manuscript.

Regarding the issue of stimulus concentration, the concentration of citric acid used in our study was relatively high compared to previous research. However, the decision to set the concentration of all five solutions at 0.5 M was made to ensure consistency in the experimental variables and to take into account the taste thresholds of all participants. These thresholds can vary significantly between individuals due to factors such as age, lifestyle habits, and genetic diversity. Moreover, no participant reported aversion or discomfort throughout the experiment.

To evaluate the impact of the level of concentration on the experimental results, we have employed taste stimulation solutions with a concentration gradient. The results demonstrate that our method and algorithm exhibit excellent classification performance even at the lower concentrations adopted in previous studies (reference 31 and 32).

Supplementary Figure 39. Effects of taste stimuli solutions with different concentrations on decoding performance. Comparison of the decoding accuracy from four groups of taste stimuli solutions with different concentrations. Noted that the concentration of 0.039 M was adopted in the reference 31 and 32 for citric acid and 0.5 M was used in our manuscript.

[5] Also the authors did not adequately explain how they differentiated taste signals from the many mechanosensory signals that arise from the tongue.

We thank the reviewer for the valuable comment. After performing bad-channel removal and power-frequency notching, we applied a common average reference to the acquired data, which is a standard technique for reducing shared common-mode noise in 256-channel data¹⁶. Additionally, since bioelectrical artefacts typically occur in higher frequency ranges, while mechanical motion artefacts generally exhibit frequency spectra below 10 Hz¹⁷, we have employed algorithmic post-processing interventions to separate the contributions of these overlapping components based on their characteristic frequencies¹⁸. Specifically, in our study, we utilized finite impulse response (FIR) filters with upper and lower cutoff frequencies of 10 Hz and 500 Hz, to minimize the influence of both bioelectrical and mechanical artefacts, thereby capturing valuable electrophysiological signals relevant to gustatory stimuli.

On the other hand, the tongue is densely populated with taste receptors, which can be activated by chemical stimulation, thereby generating electrical activity. Simultaneously, excitation of the tongue muscles leads to potential changes due to the conduction and diffusion of action potentials along the muscle fibers. Consequently, the electrical signals recorded through tongue electrodes represent a superposition of various cellular and neural electrical activities. These combined signals form the foundation for taste decoding from the electrophysiological responses of reconstructed tongue.

[6] Although the prosthesis does provide a detailed map of the tongue in response to the different taste qualities and could be useful for certain applications, it does not reach the level of significance that we would expect of a Nature Communications manuscript. I think it would

be more appropriate in a specialty journal, perhaps an engineering journal.

We thank the reviewer for consideration and valuable comments and we have significantly revised the manuscript to address the concern raised by the reviewer. We would like to highlight that this prosthesis not only provides a novel electrical mapping across the entire tongue surface to aid in assessing the surgical procedure with a new safe margin and the flap viability for postoperative monitoring and timely intervention, but more importantly offers an innovative approach for interpreting gustatory information from the reconstructed tongue flap without taste buds. Therefore, our findings may offer the possibility of discriminating flavors from unknown foods for tongue cancer patients with dysgeusia who have undergone extensive tongue flap resection, which holds significant clinical value for both research and practical application based on existing clinical studies and extensive experience in oral and maxillofacial surgery. We believe that the revised manuscript can be of immediate interest to a broad readership of Nature Communications.

References

1. Moxon, K. A. & Foffani, G. Brain-machine interfaces beyond neuroprosthetics. *Neuron* **86**, 55–67 (2015).
2. Lebedev, M. A. & Nicolelis, M. A. Brain-machine interfaces: From basic science to Neuroprostheses and neurorehabilitation. *Physiological Reviews* **97**, 767–837 (2017).
3. Moses, D. A. *et al.* Neuroprosthesis for decoding speech in a paralyzed person with anarthria. *New England Journal of Medicine* **385**, 217–227 (2021).
4. Willett, F. R. *et al.* A high-performance speech neuroprosthesis. *Nature* **620**, 1031–1036 (2023).
5. Oxley, T. J. *et al.* Motor neuroprosthesis implanted with neurointerventional surgery improves capacity for activities of daily living tasks in severe paralysis: First in-human experience. *Journal of NeuroInterventional Surgery* **13**, 102–108 (2020).
6. Sawyer, A., Cooke, L., Ramsey, N. F. & Putrino, D. The Digital Motor Output: A Conceptual Framework for a meaningful clinical performance metric for a motor neuroprosthesis. *Journal of NeuroInterventional Surgery* **16**, 443–446 (2023).
7. Togni, L. *et al.* Treatment-related dysgeusia in oral and oropharyngeal cancer: A comprehensive review. *Nutrients* **13**, 3325 (2021).
8. Tomita, S., Terao, Y., Hatano, T. & Nishimura, R. Subtotal glossectomy preserving half the tongue base prevents taste disorder in patients with tongue cancer. *International Journal of Oral and Maxillofacial Surgery* **43**, 1042–1046 (2014).
9. Elfring, T. T., Boliek, C. A., Seikaly, H., Harris, J. & Rieger, J. M. Sensory outcomes of the anterior tongue after lingual nerve repair in oropharyngeal cancer. *Journal of Oral Rehabilitation* **39**, 170–181 (2011).
10. Yuan, Y., Zhang, P., He, W. & Li, W. Comparison of oral function: Free anterolateral thigh perforator flaps versus vascularized free forearm flap for reconstruction in patients undergoing glossectomy. *Journal of Oral and Maxillofacial Surgery* **74**, (2016).
11. Fang, Q.-G. *et al.* Assessment of the quality of life of patients with Oral Cancer after pectoralis major myocutaneous flap reconstruction with a focus on speech. *Journal of Oral and Maxillofacial Surgery* **71**, (2013).

12. Airoidi, M. *et al.* Functional and psychological evaluation after flap reconstruction plus radiotherapy in oral cancer. *Head & Neck* **33**, 458–468 (2011).
13. Ecarma, M. J. & Nolden, A. A. A review of the flavor profile of metal salts: Understanding the complexity of metallic sensation. *Chemical Senses* **46**, (2021).
14. Lawless, H. T., Rapacki, F., Horne, J. & Hayes, A. The taste of calcium and magnesium salts and anionic modifications. *Food Quality and Preference* **14**, 319–325 (2003).
15. Schiffman, S. S. *et al.* The effect of sweeteners on bitter taste in young and elderly subjects. *Brain Research Bulletin* **35**, 189–204 (1994).
16. Ludwig, K. A. *et al.* Using a common average reference to improve cortical neuron recordings from microelectrode arrays. *Journal of Neurophysiology* **101**, 1679–1689 (2009).
17. Fratini, A., Cesarelli, M., Bifulco, P. & Romano, M. Relevance of motion artifact in electromyography recordings during Vibration treatment. *Journal of Electromyography and Kinesiology* **19**, 710 – 718 (2009).
18. Yin, J., Wang, S., Tat, T. & Chen, J. Motion Artefact Management for soft bioelectronics. *Nature Reviews Bioengineering* **2**, 541–558 (2024).